# Whole proteome analysis of human tankyrase knockout cells reveals targets of tankyrase-mediated degradation

Amit Bhardwaj[1], Yanling Yang[2], Beatrix Ueberheide[2] & Susan Smith[1]

Tankyrase 1 and 2 are poly(ADP-ribose) polymerases that function in pathways critical to cancer cell growth. Tankyrase-mediated PARylation marks protein targets for proteasomal degradation. Here, we generate human knockout cell lines to examine cell function and interrogate the proteome. We show that either tankyrase 1 or 2 is sufficient to maintain telomere length, but both are required to resolve telomere cohesion and maintain mitotic spindle integrity. Quantitative analysis of the proteome of tankyrase double knockout cells using isobaric tandem mass tags reveals targets of degradation, including antagonists of the Wnt/β-catenin signaling pathway (NKD1, NKD2, and HectD1) and three (Notch 1, 2, and 3) of the four Notch receptors. We show that tankyrases are required for Notch2 to exit the plasma membrane and enter the nucleus to activate transcription. Considering that Notch signaling is commonly activated in cancer, tankyrase inhibitors may have therapeutic potential in targeting this pathway.

[1] Kimmel Center for Biology and Medicine at the Skirball Institute, Department of Pathology, New York University School of Medicine, New York, NY 10016, USA. [2] Proteomics Laboratory, Department of Biochemistry and Molecular Pharmacology, New York University School of Medicine, New York, NY 10016, USA. Correspondence and requests for materials should be addressed to S.S. (email: susan.smith@med.nyu.edu)

Tankyrases function in cellular pathways that are critical to cancer cell growth including telomere cohesion and length homeostasis, Wnt/β-catenin signaling, and mitotic progression[1, 2]. Tankyrase 1 belongs to a poly(ADP-ribose) polymerase (PARP) group of enzymes that include PARP-1, 2, and 3; V-PARP; and tankyrase 1 and 2, which use $NAD^+$ as a substrate to generate ADP-ribose polymers on protein acceptors[3, 4]. PARP-1 is critical for repair of specific DNA lesions and its inhibition sensitizes cells to DNA-damaging agents[5]. Highly selective and potent inhibitors of PARP1 are currently in clinical trials for cancer[6, 7]. The preliminary success of these drugs has led to an interest in targeting other members of the PARP family. Tankyrases are overexpressed in multiple cancers and a range of potent and highly selective small molecule inhibitors of tankyrases have recently been developed[2, 8]. Elucidation of tankyrase

function in human cells will provide insights into the clinical utility of tankyrase inhibitors.

Tankyrases 1 and 2 are closely related proteins encoded by distinct genes[1]. They have a similar primary structure that includes an ankyrin repeat domain, a sterile alpha motif (SAM), and a C-terminal catalytic PARP domain[9]. The ankyrin repeats form five conserved ANK repeat clusters (ARCs) that serve as docking sites for tankyrase targets[10]. The tankyrase binding site recognized by the ARCs was initially identified as a six amino acid RxxPDG motif[11] that (through experimental approaches and sequence analysis) was extended to a maximum of eight amino acids: Rxx(small hydrophobic amino acids/G)(D/E, in addition to a small selection of other tolerated amino acids)G(no P)(D/E)[12]. A combined approach utilizing ARC crystal structures, mutagenesis, and an extensive peptide library, led to an in silico prediction of 257 potential tankyrase binding partners[12].

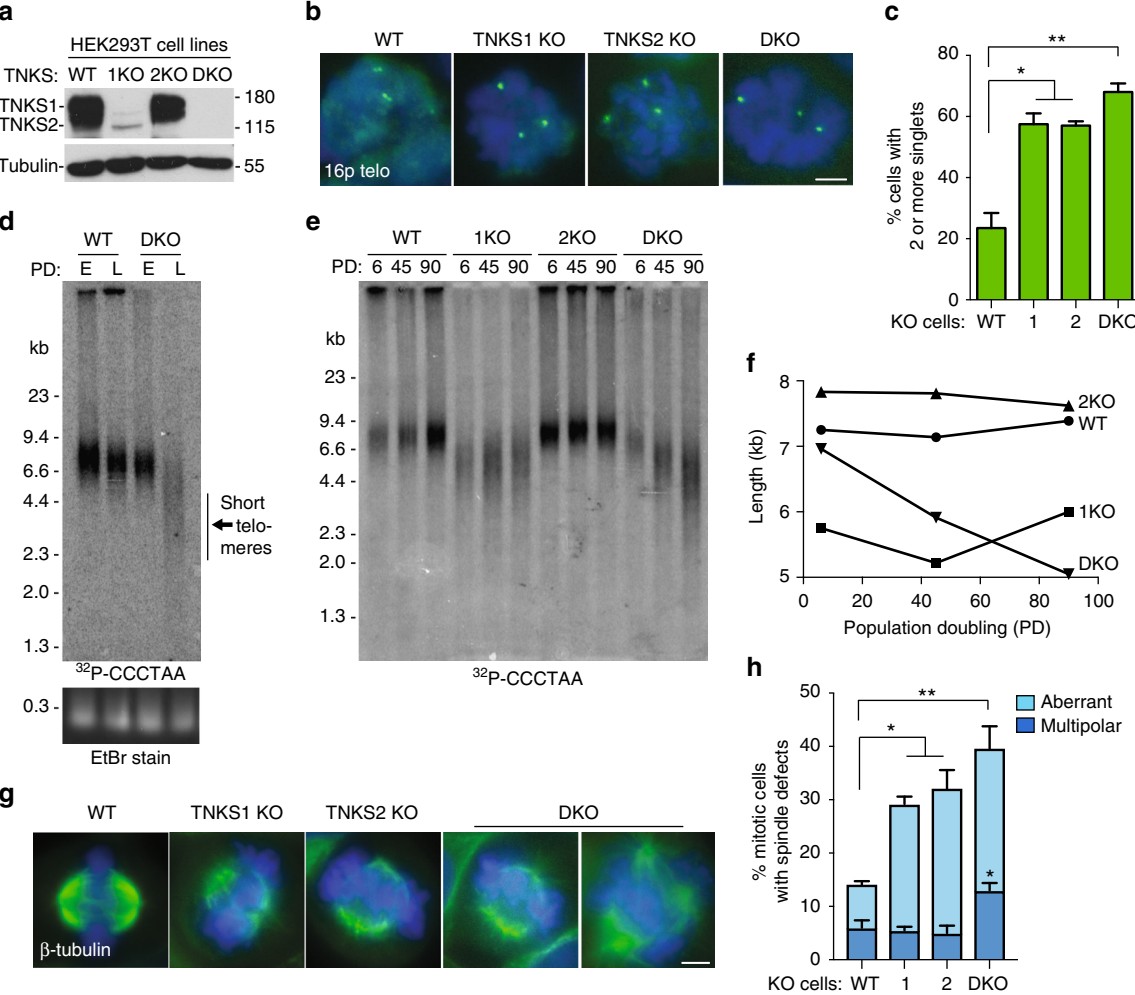

**Fig. 1** Generation and functional analysis of human tankyrase knockout cell lines. **a** Immunoblot analysis of whole cell lysates from HEK293T knockout (KO) cell lines generated by CRISPR-Cas9 stained with antibody that detects tankyrases 1 and 2. **b** FISH analysis of mitotic HEK293T WT, 1KO, 2KO, or DKO mitotic cells with a 16p telo probe (green). HEK293T cells are trisomic for chromosome 16. DNA was stained with DAPI. Scale bar, 5 μm. **c** Quantification of the frequency of mitotic cells with cohered telomeres. Average of two independent experiments ($n = 80$ cells) ± SEM. *$p \leq 0.05$, **$p \leq 0.01$, students unpaired $t$-test. **d** Analysis of telomere restriction fragments isolated from HEK293T WT or TNKS DKO cells isolated from an early (E) time immediately following the CRISPR-Cas9 screen and at a late (L) time point following passaging for ~2 months, fractionated on agarose gel, denatured, and hybridized with a $^{32}$P-[CCCATT]$_3$ probe. EtBr stain of total DNA is below. **e** Analysis of telomere restriction fragments isolated from HEK293T WT, 1KO, 2KO, or DKO cells isolated from population doubling (PD) 6, 45, or 90, fractionated on agarose gel, denatured and hybridized with a $^{32}$P-[CCCATT]$_3$ probe. **f** Graphical representation of the mean telomere length determined using Telometric (Fox Chase Cancer Center). **g** Immunofluorescence analysis of HEK293T WT, 1KO, 2KO, or DKO cells following fixation with 2% paraformaldehyde and stained with β-tubulin antibody (green) and DAPI (blue). Scale bar, 5 μm. **h** Quantification of the frequency of cells with defective mitotic spindles. Average of two independent experiments ($n = 66$–71 mitotic cells each) ± SEM. *$p \leq 0.05$, **$p \leq 0.01$, Student's unpaired $t$-test

Tankyrase 1, due to its greater abundance and easy detection, is the best studied of the two tankyrase isoforms. Depletion analysis in human cells has revealed functions at telomeres, mitotic spindles, and in Glut4 vesicle trafficking[1, 2]. Whether tankyrase 2 can substitute for tankyrase 1 or if it has distinct functions has not been determined. Knockout of tankyrase 1 or 2 in mice revealed only minor phenotypes[13–15], however the double knockout was embryonic lethal, indicating functional redundancy[13]. Despite the high conservation of tankyrases between mouse and human[1], not all tankyrase functions are conserved. For example, the TRF1 tankyrase-binding site RGCADG is deleted in mouse and as a result, tankyrase does not bind mouse TRF1[11] or go to telomeres in mouse cells[16], hence the telomeric function (and potentially other functions) of tankyrases may be unique to human cells[1, 17].

Insight into the potential for small-molecule inhibitors of tankyrases in cancer came to light following a chemical genetic screen for inhibitors of the Wnt/β-catenin signaling pathway, which is activated in many cancers[18]. Wnt controls the stability of the transcriptional coactivator β-catenin. In the absence of the Wnt signal, a cytoplasmic "β-catenin destruction complex" containing the key concentration-limiting component Axin, APC (adenomatous polyposis coli), CK1α, and GSK3β, promotes degradation of β-catenin. Upon Wnt activation, the β-catenin destruction complex is inactivated by the cytoplasmic transducer Disheveled (DVL), leading to increased β-catenin protein that then enters the nucleus to activate transcription[18, 19]. The screen identified XAV939, a small molecule inhibitor of tankyrases and further demonstrated that tankyrases control the stability of Axin[20]. Tankyrase-mediated PARylation of axin results in its K48-linked polyubiquitination and proteasomal degradation, thereby stabilizing β-catenin and promoting cancer cell growth[20].

Ubiquitylation of PARylated targets (including tankyrases) is mediated by the PAR-binding E3 ligase RNF146[21–23]. Over the last few years, five more targets were identified: 3BP2 (c-ABL SH3 domain binding protein 2)[24]; BLZF1 (basic leucine zipper factor 1)[23]; CASC3 (cancer susceptibility factor 3)[23]; PTEN (phosphatidylinositol (3,4,5)-trisphosphate phosphatase and tensin homolog deleted from chromosome 10), a critical tumor suppressor[25]; and AMOT (Angiomotin), a regulator of YAP (Yes-associated protein), a component of the HIPPO signaling pathways that is overexpressed in various cancers[26]. The total number and range of targets remain to be determined.

To elucidate the functions of tankyrases in human cells, we generated single and double tankyrase knockout human cell lines using CRISPR-Cas9. Functional analysis indicates distinct and overlapping roles for tankyrases. In addition, we performed a quantitative analysis of the proteome in tankyrase double knockout cells and report on targets of tankyrase–mediated proteasomal degradation.

## Results

**Functional analyses of tankyrase knockout cell lines**. We used CRISPR-Cas9 technology to generate tankyrase knockout (KO) human HEK293T cell lines; three TNKS1 KO; one TNKS2 KO; and one TNKS1/TNKS2 KO clones were generated (Supplementary Figure 1 a, b). Tankyrase protein was analyzed in the KO cell lines using immunoblot analysis with antibodies raised against tankyrase 1 (that also cross-react weakly with tankyrase 2)[15]. Figure 1a shows loss of tankyrase 1 in TNKS1 KO (1KO), loss of tankyrase 2 in TNKS2 KO (2KO), and loss of both in the double knockout TNKS1 KO/TNKS2 KO (DKO).

Tankyrase 1 is required for resolution of sister telomeres prior to mitosis[27]. The role of tankyrase 2 in this process has not been determined. To analyze telomere cohesion, we isolated cells by mitotic shake-off and subjected them to FISH analysis with a sub telomere specific probe 16p (Fig. 1b, c). In wild-type cells telomeres are resolved in mitosis and appear as doublets. TNKS1 KO cells show unresolved cohesion (singlets). Surprisingly, TNKS2 KO showed the same persistent telomere cohesion phenotype as TNKS1 KO, indicating that tankyrase 2 is required to resolve telomere cohesion. TNKS1/TNKS2 DKO cells revealed an even greater level (small, but significant), of persistent cohesion than the single KOs. Thus, at endogenous levels tankyrases 1 and 2 are each required and one cannot compensate for the other. To perform rescue analysis, we generated stable DKO cell lines overexpressing tankyrase 1, tankyrase 2, or both at levels >10-fold relative to the endogenous proteins (Supplementary Figure 2a). Overexpression of tankyrase 1 or 2 was unable to rescue persistent telomere cohesion; this required overexpression of both tankyrase 1 and 2 (Supplementary Figure 2b), consistent with each being required.

Overexpression of tankyrase 1 or 2 in the nucleus induced telomere lengthening, indicating tankyrase as a positive regulator of telomere length[28, 29]. Depletion of tankyrase 1 led to minor transient telomere shortening[17]. To determine whether knockout of tankyrases results in telomere shortening, we sub cultured WT and DKO cells side by side for several months and performed telomere restriction fragment analysis on early vs. late population doublings (PD). As shown in Fig. 1d, we observed telomere shortening in DKO cells. To determine the contribution of the individual tankyrases, we carried the four cell lines WT, 1KO, 2KO, and DKO side-by-side and performed telomere restriction fragment analysis on cells from PD 6, 45, and 90. As shown in Fig. 1e, f, either tankyrase 1 or 2 was sufficient to maintain telomere length, whereas the DKO exhibited telomere shortening. Quantitative-RT-PCR showed no change in levels of telomerase (hTERT) RNA (Supplementary Figure 2c). Note that since the lines were isolated from single-cell clones, the starting length varies for each isolate. Nonetheless, only the DKO exhibits telomere shortening. Analysis of two additional TNKS1 KO lines showed no telomere shortening when carried for over 100 PD (Supplementary Figure 2d). Overexpression (>10-fold) of either tankyrase 1 or 2 in DKO cells rescued telomere shortening (Supplementary Figure 2e), consistent with the observation that either tankyrase 1 or 2 can maintain telomere length.

Tankyrase 1 localizes to spindle poles in mitosis through its target the nuclear mitotic apparatus protein NuMA[30, 31] and is required for assembly of normal bipolar spindles at mitosis[30, 32]. The role of tankyrase 2 in this process has not been determined. To analyze mitotic spindle integrity, we subjected the TNKS KO cell lines to immunofluorescence analysis with β-tubulin antibody. As shown in Fig. 1g and h, the 1KO and 2KO cell lines showed similar levels of defective mitotic spindles, indicating that tankyrases 1 and 2 are each required for mitotic spindle integrity. The DKO cells showed a similar level of aberrant spindles as the single knockouts, as well as a two-fold increase in mitotic cells with multipolar spindles. Overexpression of tankyrase 1 or 2 at levels >10-fold relative to the endogenous proteins was unable to rescue spindle defects; this required overexpression of both tankyrases 1 and 2 (Supplementary Figure 2f), consistent with each being required.

**Quantitative analysis of the proteome of tankyrase DKO cells**. Tankyrase-mediated PARylation of Axin results in K48-linked ubiquitylation and proteasomal degradation, and subsequent stabilization of β-catenin[20]. We used immunoblot analysis to determine whether proteins in the Wnt/β-catenin pathway were affected in the TNKS KO cell lines. As shown in Fig. 2a, Axin1 was increased and β-catenin decreased in 1KO, 2KO, and DKO cells. We observed the most robust effect (stabilization of Axin1

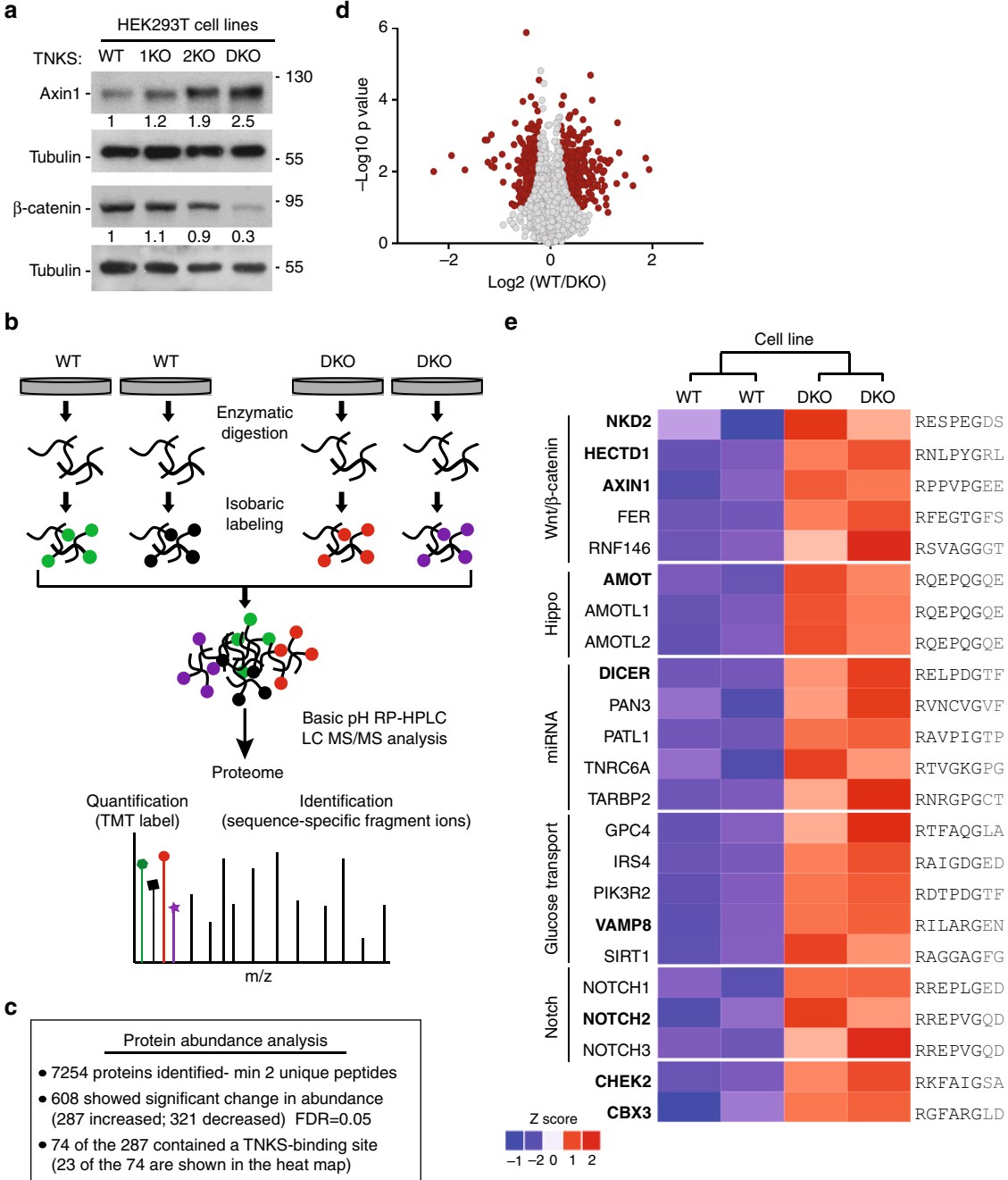

**Fig. 2** Quantitative proteomic analysis of TNKS DKO cells. **a** Immunoblot of HEK293T TNKS KO cell lines demonstrating stabilization of Axin1 and reduction of β-catenin in TNKS KO cell lines. Protein levels of Axin and β-catenin in total cell lysates from KO cell lines relative to tubulin and normalized to WT cells are indicated below the blot. **b** Experimental strategy for analyzing protein abundance in TNKS DKO vs. WT cells using TMT labeling. **c** Results of protein abundance analysis. **d** Volcano plot indicating the 608 proteins showing a significant change in abundance (see Supplementary Data 1). **e** Heat map indicating 23 of the 74 proteins that are stabilized in TNKS DKO cells with a tankyrase binding site indicated on the right; additional sites (if present) are noted in Supplementary Data 2. Note, Axin1 has a non-canonical site, RxxVxGxE. Gene names in bold were selected for further analysis

and loss of β-catenin) in the DKO cell line, indicating it as the optimal line to use for identifying targets of tankyrase-mediated degradation.

We performed a quantitative analysis of the proteome of DKO vs. wild-type cells using isobaric tag based TMT (tandem mass tag) labeling and LC-MS/MS[33–36] (see schematic in Fig. 2b). Protein extracts were prepared from two biological replicates of each cell line (DKO and WT). After cell lysis and proteolytic digestion each sample was individually covalently labeled with a distinct isobaric tag. The samples were then combined and

fractionated into 30 final fractions using basic RP-HPLC[37, 38] and interrogated by mass spectrometry. We quantified 7254 proteins (Fig. 2c; Supplementary Data 1, sheet 1) of which 608 showed significant changes in abundance in the DKO (Supplementary Data 1, sheet 2) as determined by Welch's $T$-test filtered for 5% false discovery rate (FDR) using Benjamini–Hochberg (Fig. 2d). Those proteins with greater than 1.5-fold change in abundance are indicated in Supplementary Data 1, sheet 3. Of the 608 proteins, 287 showed an increase and 321 a decrease in abundance.

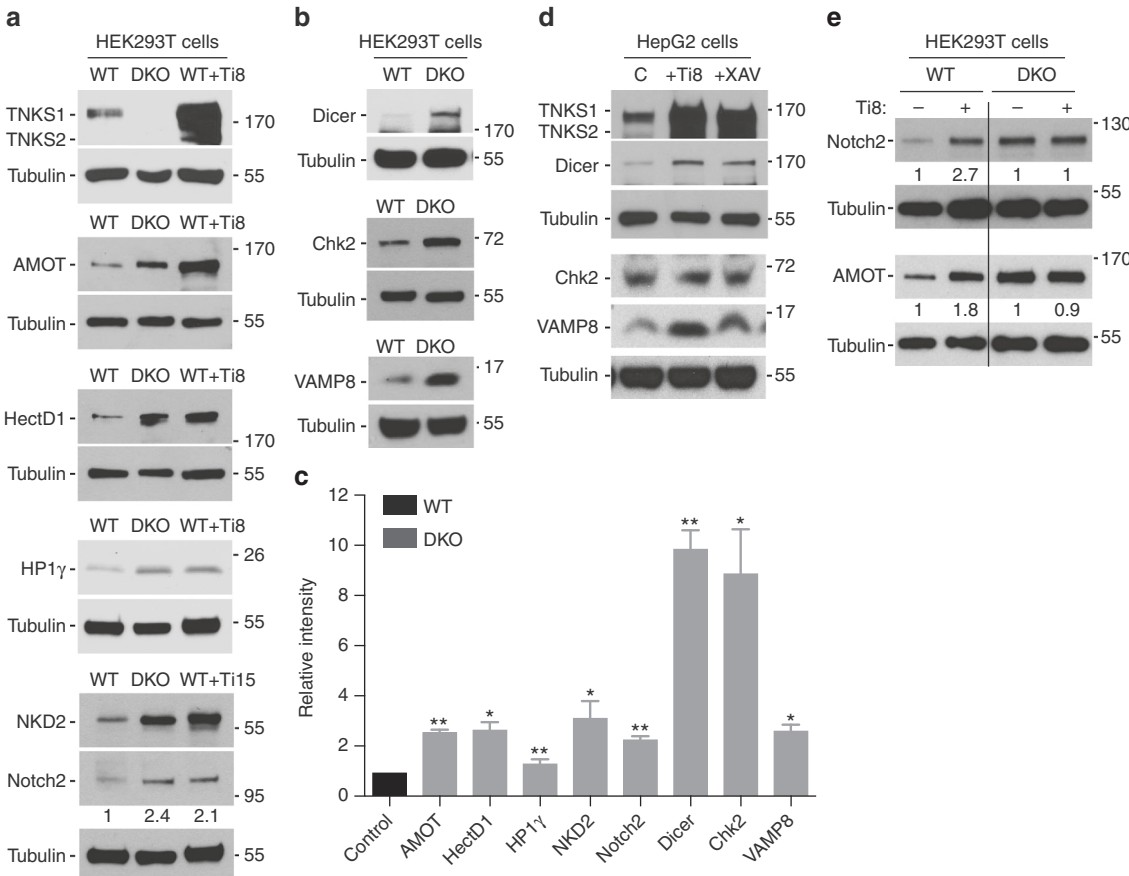

**Fig. 3** Validation of candidates from the proteomic screen. **a** Immunoblot analysis of HEK293T WT, TNKS DKO, or WT plus tankyrase inhibitor (Ti8 or Ti15) with the indicated antibodies. Protein levels of Notch2 in DKO and WT + Ti15 relative to tubulin and normalized to WT cells are indicated below the blot. **b** Immunoblot analysis of HEK293T WT or TNKS DKO cells with the indicated antibodies. **c** Plot of the abundance of the indicated proteins in TNKS DKO relative to WT cells. Average of two or more independent experiments ± SEM. *$p \le 0.05$, **$p \le 0.01$, Student's unpaired $t$-test. **d** Immunoblot analysis of HepG2 cells treated with tankyrase inhibitor (Ti8 or XAV939) with the indicated antibodies. **e** Immunoblot analysis of HEK293T WT or DKO cells without (−) or with (+) tankyrase inhibitor Ti8 with the indicated antibodies. Protein levels of Notch2 or AMOT in WT + Ti8 relative to tubulin and normalized to WT cells and of Notch2 or AMOT in DKO + Ti8 relative to tubulin and normalized to DKO cells are indicated below the blot

Validation of tankyrase degradation targets. In selecting candidates to focus on, we reasoned that proteins that were increased in abundance and contained a strong tankyrase consensus binding site (RXXG[P/A/C]XG) had the best chance of being direct targets of tankyrase-mediated ubiquitylation and degradation. Of the 287 proteins showing increased abundance, 74 contained a strong tankyrase consensus binding site (listed in Supplementary Data 2). Several of the 74 were previously identified as targets of tankyrase-mediated degradation: Axin1, BLZ1, RNF146, and the Angiomotin family members AMOT, AMOTL1, and AMOTL2, thereby validating our strategy.

We subjected the 74 candidate proteins to pathway analysis in Cytoscape and additional manual curation using published data to cluster the candidates into pathways. Twenty-one of the seventy-four were grouped by pathway (Fig. 2e). Five proteins fell into the Wnt/β-catenin signaling pathway including the aforementioned Axin1 and RNF146 plus three other potential targets of tankyrase. Interestingly, all three have been reported to (like Axin) negatively regulate β-catenin. HectD1 (HECT Domain E3 Ubiquitin Protein Ligase 1) is an E3 ligase that modifies the APC component of the destruction complex with K63-ubiquitin to stabilize the APC-Axin interaction and promote degradation of β-catenin[39]. NKD2 (Naked Cuticle Homolog 2) promotes degradation of DVL-1 (disheveled) to prevent it from inactivating the GSK3β component of the destruction complex and promoting

degradation of β-catenin[40]. Fer is a tyrosine kinase that phosphorylates the LRP6 transducer to negatively regulate Wnt/β-catenin signaling[41]. The three Angiomotin family members (AMOT1, AMOTL1,and AMOTL2) of the HIPPO pathway have been shown to indirectly impact the Wnt/β-catenin pathway[42]. Additional pathways identified include: the microRNA processing and glucose transport pathways (each represented by five proteins) and the Notch signaling pathway (containing three of the four Notch receptors) (Fig. 2e). The remaining candidates distributed across a wide range of pathways represented by only 1 or 2 proteins.

For validation, we selected seven candidates (plus the known AMOT) from these groups as well as two additional candidates that we reasoned might be involve in tankyrase's telomeric functions: *CBX3* (HP1ϒ, a heterochromatin protein required for telomere cohesion[43]) and *CHEK2* (Chk2, required for the DNA damage response) (Fig. 2e; proteins selected for analysis are indicated in bold). Immunoblot analysis of extracts from HEK293T WT vs. DKO cells revealed an increase in AMOT, HectD1, HP1ϒ, NKD2, and Notch2 protein levels (Fig. 3a). Treatment with tankyrase-specific inhibitors compound 8 (Ti8) or 15 (Ti15)[44] induced a similar increase in protein levels (Fig. 3a). Additional immunoblot analysis showed an increase in Dicer, Chk2, and VAMP8 protein levels in HEK293T DKO vs. WT cells (Fig. 3b). Overall, we found that the seven candidates

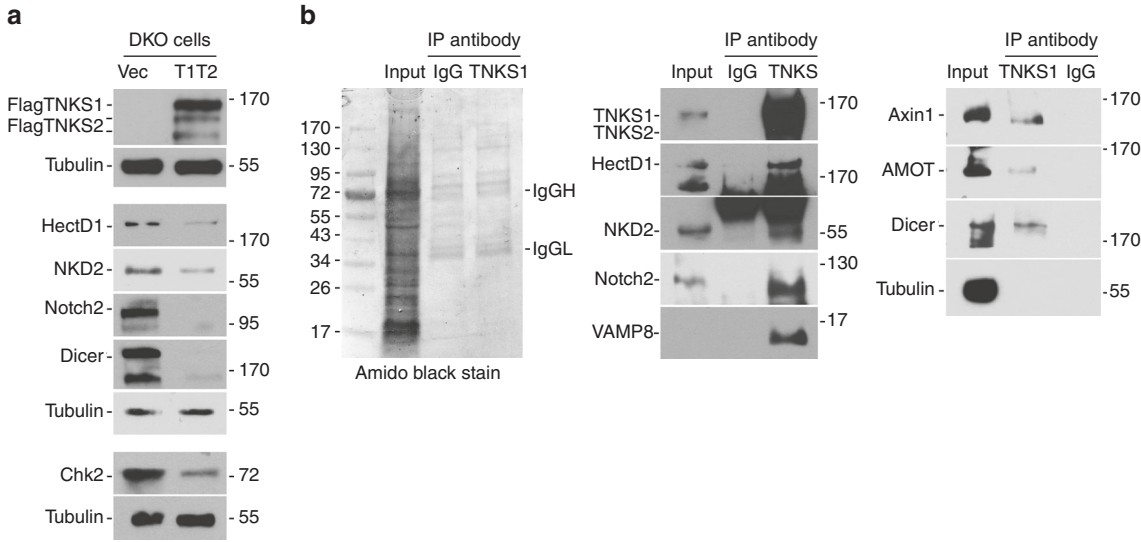

**Fig. 4** Rescue and immunoprecipitation analysis of candidates for the proteomic screen. **a** Reintroduction of tankyrases into DKO cells induces loss of candidate proteins. Immunoblot analysis of HEK293T DKO cells stably expressing a vector control or Flag-TNKS1 and Flag-TNKS2 (T1T2) with the indicated antibodies. **b** Coimmunoprecipitation of candidate proteins with tankyrase. Immunoblot analysis of HEK293T cells immunoprecipitated with control or TNKS IgG, stained with amido black (left panel) and probed with the indicated antibodies (middle and right panels)

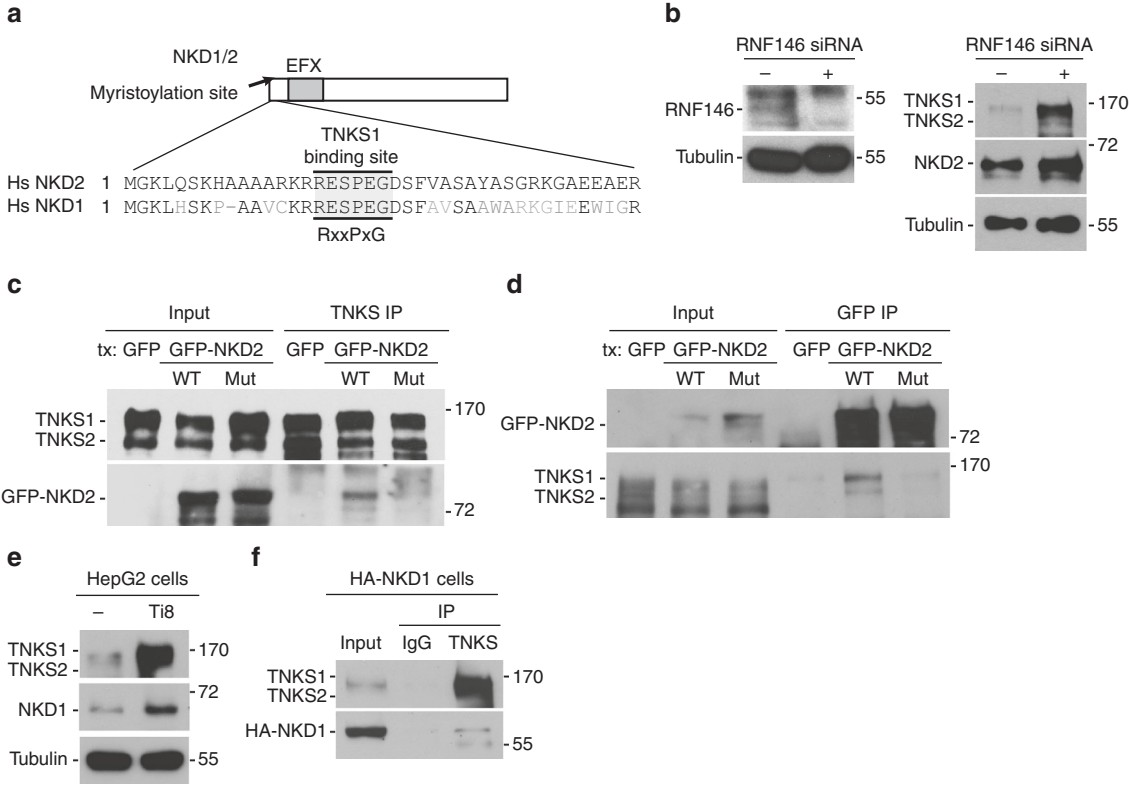

**Fig. 5** NKD1 and NKD2 are targets of tankyrase. **a** Schematic diagram of NKD1/2 showing the N-terminal myristoylation site. The shaded box indicates the conserved EFX (EF-hand containing) domain that binds DVL. Alignment of the tankyrase-binding site in human NKD1 and NKD2. Identical amino acids are in black. **b** Immunoblot analysis showing NKD2 is stabilized in RNF146 siRNA-treated HEK293T cells. **c**, **d** Endogenous tankyrase and GFP-NKD2 coimmunoprecipitate, dependent on the NKD2 tankyrase-binding site. Immunoblot analysis of HEK293T cells transfected with GFP, GFP-NKD2 WT, or GFP-NKD2 Mut, immunoprecipitated with **c** tankyrase or **d** GFP antibodies, and probed with the indicated antibodies. **e** Immunoblot analysis showing that NKD1 is stabilized upon treatment of HepG2 cells with tankyrase inhibitor Ti8. **f** Immunoblot analysis showing that HA-NKD1 coimmunoprecitates with tankyrase. Immunoblot of HEK293T cells transfected with HA-NKD1, immunoprecipitated with control or TNKS IgG, and probed with the indicated antibodies

tested showed a statistically significant increase in protein levels in DKO vs. WT HEK293T cells (Fig. 3c). To determine whether stabilization could be detected in a different cell line and with different inhibitors, we subjected HepG2 cells to tankyrase inhibitor treatment (Ti8 or XAV939) and observed an increase in Dicer and VAMP8, but not Chk2 (Fig. 3d), which unlike the other targets did not show stabilization with tankyrase inhibitor treatment. Finally, to determine whether stabilization of targets by inhibitor treatment was due to tankyrases we compared Ti8 inhibitor-treated WT vs. DKO cells side-by-side. As shown in Fig. 3e, Ti8 induced stabilization of AMOT and Notch2 in WT, but had no effect in DKO cells.

To further validate that candidates were targets of tankyrases we measured protein levels in DKO cells expressing vector vs. tankyrase 1 and tankyrase 2 (T1T2). Immunoblot analysis shows that all proteins tested (HectD1, NKD2, Notch2, Dicer, and Chk2) were reduced in the tankyrase rescued line compared to vector control (Fig. 4a). Next we asked whether the candidates interacted with tankyrase by performing immunoprecipitation analysis on endogenous proteins in HEK293T WT cells. Cell lysates were immunoprecipitated with control or TNKS1 IgG and analyzed by immunoblot. As shown in Fig. 4b, HectD1, NKD2, Notch2, VAMP8, Dicer, Axin1, and AMOT each coimmunoprecipitated with tankyrase 1. HP1ϒ and Chk2 were not detected in the immunoprecipitates.

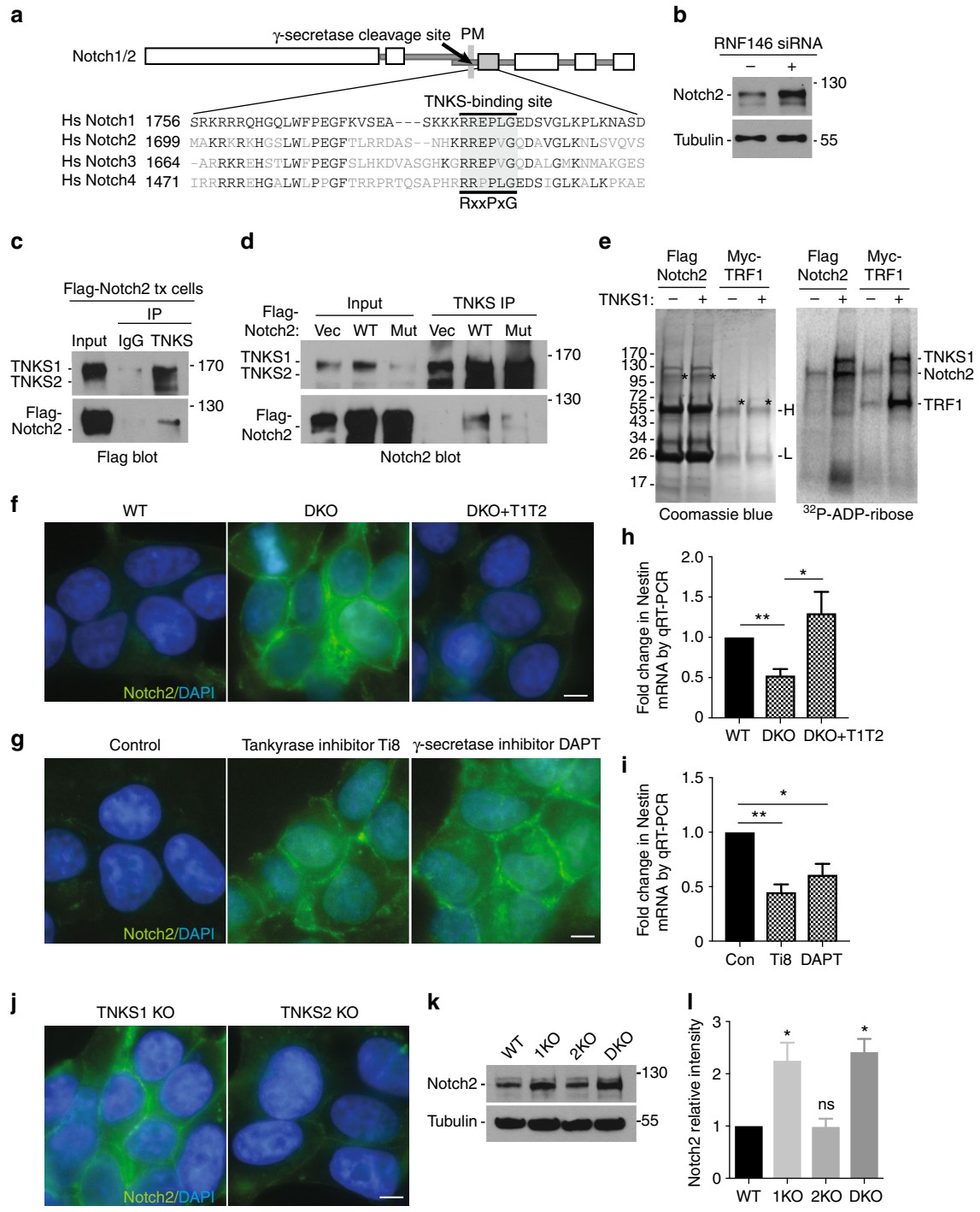

**Characterization of the tankyrase target NKD.** NKD2 acts as negative regulator of the Wnt/β-catenin pathway through the Disheveled (DVL) component of the pathway[40]. Upon Wnt activation, DVL transduces the signal by inactivating the destruction complex to promote accumulation of β-catenin. NKD2 antagonizes the signaling pathway by binding directly to DVL-1, preventing inactivation of the destruction complex. NKD2 has an N-terminal myristoylation site that localizes it to the plasma membrane and EF-hand-like motifs that bind directly to DVL-1[45] (see schematic diagram in Fig. 5a). The RESPEG tankyrase binding site in NKD2 is near the N-terminus, adjacent to the myristoylation site and is conserved between NKD2 human and its homolog NKD1 (Fig. 5a). In support of NKD2 as a target of tankyrase-mediated degradation, we have already shown that: NKD2 levels increased in TNKS DKO cells and in WT cells treated with a tankyrase inhibitor (Fig. 3a); reintroduction of TNKS1 and 2 into DKO cells led to a decrease in NKD2 (Fig. 4a); and endogenous NKD2 was bound to endogenous TNKS1 following immunoprecipitation analysis of HEK293T cells (Fig. 4b). Next, to determine whether NKD2 (like Axin) is targeted for degradation by the tankyrase-associated PAR-dependent E3 ligase RNF146, we analyzed NKD2 levels in cells depleted of RNF146 using siRNA. As shown in Fig. 5b, NKD2 levels were increased in RNF146-depleted cells. We next asked whether tankyrase binds to NKD2 through its RESPEG tankyrase binding motif. We mutated the terminal glycine in the tankyrase binding site to alanine (from RESPEG to RESPEA) in a GFP-tagged NKD2 allele. The GFP-NKD2 WT and mutant plasmids were transfected into HEK293T WT cells and immunoprecipitated with anti-TNKS1 antibody. As shown in Fig. 5c, GFP-NKD2 WT coimmunoprecipitated with endogenous TNKS1, whereas interaction with the GFP-NKD2 mutant was strongly reduced. Conversely, endogenous TNKS1 coimmunoprecipitated with GFP-NKD2 WT, whereas interaction with the GFP-NKD2 mutant was strongly reduced (Fig. 5d).

NKD1 shares 43% identity with NKD2 and also interacts with DVL to negatively regulate the Wnt/β-catenin pathway[40, 46]. Since the TNKS-binding site is conserved between human NKD1 and 2 (Fig. 5a), we wondered whether NKD1 is also a target of tankyrase. We did not detect NKD1 in our proteomic screen and we were unable to detect the protein by immunoblotting HEK293T cells. However, immunoblot analysis of another cell type, HEPG2, revealed expression of NKD1 protein, which was stabilized upon treatment with a TNKS inhibitor Ti8 (Fig. 5e). Immunoprecipitation analysis showed that transfected HA-NKD1 coimmunoprecipitated with endogenous TNKS (Fig. 5f). Together these data indicate NKD1 as a target of tankyrase.

**Characterization of the tankyrase target Notch.** Notch signaling functions in cell fate determination, development, and cancer[47, 48]. There are four Notch receptors (Notch1–4) in mammals, with Notch1 and 2 most closely related. Notch is a single-pass transmembrane receptor that exists as a heterodimer comprised of an extracellular ligand binding domain and an intracellular domain that mediates signaling (see schematic diagram in Fig. 6a). Upon activation by ligand, the extracellular domain is cleaved close to the membrane, leaving the intracellular portion anchored to the plasma membrane. Subsequently γ-secretase cleaves the transmembrane domain to release the Notch intracellular domain (Notch-ICD), which then enters the nucleus and binds to the RelA family transcription factor RBPJ (recombining binding protein suppressor of hairless). Interaction of the Notch-ICD through its N-terminal RAM (RPBJ association molecule) domain with RBPJ leads to displacement of repressive factors and transcriptional activation.

We identified Notch1, 2, and 3 in our proteomic screen. Interestingly, all four members of the Notch family contain a conserved RxxPxG motif in the RAM domain near the N terminus of the ICD (Fig. 6a). In support of Notch2 as a target of tankyrase-mediated degradation, we have shown thus far (by immunoblot analysis using an antibody directed against the C-terminal domain of Notch2) that endogenous Notch2 (intracellular portion) is increased in TNKS DKO cells and in WT cells treated with tankyrase inhibitor (Fig. 3a), and it is decreased upon reintroduction of tankyrases 1 and 2 into DKO cells (Fig. 4a). Immunoprecipitation analysis showed that endogenous Notch2 was bound to endogenous tankyrase in HEK293T cells (Fig. 4b). We focus below on Notch2, but we also found that Notch1 is stabilized in TNKS DKO cells (Supplementary Figure 3a) and the low levels of Notch 1 are restored by reintroduction of tankyrases 1 and 2 (Supplementary Figure 3b). For Notch3, we were unable to detect in HEK293T cells, but immunoblot analysis of MCF7 cells showed that Notch3 increased upon treatment with tankyrase inhibitor Ti8 (Supplementary Figure 3c).

To extend analysis of Notch2, we measured protein levels in RNF146-depleted cells. As shown in Fig. 6b, Notch2 was increased, indicating it as a target of the RNF146 E3 ligase. We further show that epitope-tagged Flag-Notch2-ICD coimmunoprecipitated with endogenous tankyrase (Fig. 6c). To test the dependence of the interaction on the Notch2 RxxPxG motif, we mutated the terminal glycine in the tankyrase-binding site to alanine (from RREPVG to RREPVA). As shown in Fig. 6d, Flag-Notch2-ICD WT coimmunoprecipitated with endogenous TNKS to a greater extent than the Flag-Notch2-ICD mutant. To

**Fig. 6** Notch is a target of tankyrase. **a** Schematic diagram of Notch1/2. Shaded box; RAM (RPBJ association molecule) domain that binds RPBJ (recombining binding protein suppressor of hairless). PM plasma membrane. Alignment of tankyrase-binding sites; identical amino acids in black. The acidic residue at the 8th position of the tankyrase-binding motif indicates an 8-amino-acid motif. **b** Immunoblot analysis showing Notch2 is stabilized in RNF146 siRNA-treated HEK293T cells. **c** Endogenous tankyrase and Flag-Notch2 coimmunoprecipitate. Immunoblot analysis of HEK293T cells transfected with Flag-Notch2 and immunoprecipitated with tankyrase IgG. **d** Endogenous tankyrase and Flag-Notch2 coimmunoprecipitate, dependent on the Notch2 tankyrase-binding site. Immunoblot analysis of HEK293T cells transfected with vector, Flag-Notch2 WT, or Flag-Notch2 Mut and immunoprecipitated with anti-tankyrase IgG. **e** Notch2 is a target for ADP-ribosylation by tankyrase 1. Flag-Notch2 or Myc-TRF1 immunoprecipitates from HEK293T cells were incubated with $^{32}$P-NAD$^{+}$ and recombinant tankyrase 1, fractionated by SDS-PAGE and visualized by Coomassie blue (left panel; asterisks indicate FlagNotch2 and MycTRF1) and autoradiography (right panel). **f** Notch2 is increased at the plasma membrane in TNKS DKO cells. Immunofluorescence analysis of formaldehyde-fixed HEK293T WT, TNKS DKO, or TNKS DKO expressing tankyrase 1 and 2 (T1T2) cells stained with anti-Notch2 antibodies (green). DNA was stained with DAPI (blue). Scale bar, 5 μm. **g** Notch2 is increased at the plasma membrane in HeLa cells treated with inhibitors to tankyrase (Ti8) or γ-secretase (DAPT). Immunofluorescence analysis was performed as described in **f**. **h, i** Expression of the Notch target gene Nestin is reduced in **h** TNKS DKO cells and in **i** inhibitor-treated HEK293T cells. mRNA levels were quantified using qRT-PCR. Average of two to four independent experiments (with three technical replicates each) ± SD. *$p \leq 0.05$, **$p \leq 0.01$, Student's unpaired $t$-test. **j–l** Notch2 stability is regulated by TNKS1, not TNKS2. **j** Immunofluorescence analysis of TNKS1 and TNKS2 DKO cells was performed as in **f**. **k** Immunoblot analysis of HEK293T WT and TNKS KO cells lines. **l** graphical representation of the abundance in TNKS KO cells relative to WT. Average of two independent experiments ± SEM. *$p \leq 0.05$, Student's unpaired $t$-test

determine if Notch is a target of ADP-ribosylation by tankyrase 1, we immunoprecipitated Flag-Notch2-ICD (or the known target Myc-TRF1 as a positive control) and performed a PARP assay on the immunoprecipitates using a $^{32}$P-NAD$^+$ substrate and recombinant tankyrase 1. As shown in Fig. 6e, Notch2 is ADP-ribosylated in vitro by tankyrase 1.

To evaluate Notch2 at the cellular level, we performed immunofluorescence analysis of endogenous Notch2 in wild-type vs. DKO HEK293T cells. As shown in Fig. 6f, we detected weak localization of Notch2 to the plasma membrane in wild-type cells that was dramatically increased in DKO cells and reduced back to wild-type levels upon reintroduction of tankyrases. The Notch2 staining pattern in the DKO cells was reminiscent of that observed for a Notch1 reporter allele (Notch1ΔE-eGFP) in HeLa cells following treatment with the γ-secretase inhibitor DAPT, which blocks membrane cleavage and generation of the Notch-ICD; the Notch1 reporter accumulated at the plasma membrane rather than in the nucleus[49]. To determine whether the pattern of localization was similar, we analyzed Notch2 localization in HeLa cells treated with tankyrase inhibitor Ti8 or γ-secretase inhibitor DAPT. As shown in Fig. 6g, inhibition of tankyrases or γ-secretase led to accumulation of Notch2 at the plasma membrane. A similar accumulation of Notch3 at the plasma membrane was observed in MCF7 cells treated with Ti8 (Supplementary Figure 3d). Immunoblot analysis showed that DAPT (like Ti8) led to stabilization of the intracellular portion of Notch2 (Supplementary Figure 3e).

We next sought to determine whether it was the cleaved ICD or uncleaved membrane-bound form of Notch2 that was increased following inhibition of tankyrases. Our immunoblot analysis thus far (using antibody directed against the C terminus of Notch2) did not distinguish between the two forms, since they differ only slightly in molecular weight. We thus performed the analysis using antibody that detects only the cleaved (Val1697) form of Notch2[50]. As shown in Supplementary Figure 3f, we observed a reduction in cleaved Notch2 upon Ti8 or DAPT treatment, indicating that the observed increase in Notch2 upon tankyrase inhibition is due to an increase in the membrane-bound uncleaved form. Immunoprecipitation analysis shows that tankyrase can bind to Notch2 in DAPT-treated cells (Supplementary Figure 3g).

Our studies indicate that tankyrases (like γ-secretase) are required to generate the activated Notch2-ICD. One prediction is that transcription of Notch2 targets would be diminished in the absence of tankyrases. For one such target Nestin[51], we observed a significant decrease in protein level in TNKS DKO cells (see Supplementary Data 1, sheet 2). To determine whether this decrease occurred at the mRNA level, we performed quantitative mRNA analysis. As shown in Fig. 6h, we observed significant reduction of Nestin mRNA levels in DKO cells that was rescued by reintroduction of tankyrases. A similar decrease in Nestin mRNA was observed upon Ti8 or DAPT treatment (Fig. 6i).

Finally, we asked if Notch2 interaction was specific for tankyrase 1 or 2. Immunofluorescence analysis of the single KO TNKS HEK293T cell lines revealed an increase in Notch2 staining in TNKS1 KO but not TNKS2 KO (Fig. 6j) that was confirmed by immunoblot analysis (Fig. 6k, l), indicating specificity (at least in the case of Notch2) for tankyrase 1.

## Discussion

In this study, we have elucidated the contribution of tankyrase 1 vs. tankyrase 2 in telomere cohesion, telomere length regulation, mitotic spindle integrity, and protein degradation. We found that while either tankyrase 1 or 2 is sufficient to maintain telomere length, both tankyrases 1 and 2 are required to resolve sister telomere cohesion and to maintain mitotic spindle structure. Our observation that overexpression of tankyrase 1 or 2 at levels >10-fold relative to the endogenous proteins was unable to rescue persistent cohesion or mitotic spindle integrity argues against the idea that the total amount of tankyrases 1 and 2 is required for function and instead suggests a functional difference between tankyrases and/or a putative mechanism that requires collaboration. One might anticipate that unique requirements would be mediated by distinct binding partners. However, thus far all tankyrase binding partners use their RxxG(P/A/C)xG motif to bind to the ankyrin repeats of either tankyrases 1 and 2. Perhaps yet to be identified partners will bind to other less conserved domains of the individual tankyrases to execute distinct functions. Additionally, some functions of tankyrase may require that it acts as a heteropolymer, since tankyrases can self-associate into heterotypic oligomers[52–54]. Our observation that the effect on Notch2 stability and localization occurs in TNKS1 (but not TNKS2) KO cells provides an example of a protein target (Notch2) showing specificity for one of the tankyrases (tankyrase 1) in living cells. It will be interesting to determine here if tankyrase 1 has a distinct subcellular localization or unique protein binding partner that promotes association with Notch2.

Interestingly, despite the fact that the TNKS double knockout leads to embryonic lethality in mice[13], human cells (at least HEK293T) can survive without tankyrases. The reason for the lethality in mice is not known, however considering that tankyrase targets are found in important developmental pathways such as Wnt/β-catenin and notch signaling, tankyrase deletion could disrupt signaling pathways that are essential during early development of the whole organism. Such disruption may have less of an effect on cells grown in culture. Additionally, cells (particularly cancer cells) may find a way to compensate for the loss of tankyrase during clonal outgrowth. Considering the potential use of tankyrase small-molecule inhibitors in anti-cancer therapy, it is interesting to see that cancer cells can survive without tankyrases. Thus, anti-cancer strategies would likely require combinational approaches, similar to PARP inhibitors that are used in combination with DNA-damaging agents or in cells defective in homologous recombination. Since tankyrase knockout results in telomere shortening, inhibitors could be used in combination with telomerase inhibitors[55]. In another example, since tankyrase knockout results in defective mitotic spindles, inhibitors could be used in conjunction with mitotic inhibitors or spindle poisons to kill cancer cells.

We performed an unbiased quantitative proteomic screen in WT vs. DKO cells and identified 74 putative tankyrase-binding site containing candidate targets of tankyrase-mediated degradation. Six of the seventy-four were previously identified and we confirmed seven additional candidates, thereby giving confidence in our strategy. While some of the candidates could be pooled into small functional groups, many represented an array of unique functions. This is perhaps not surprising, as tankyrases localizes throughout the cell to multiple compartments and exhibit a broad range of functions. The identification of more targets will likely provide insights into tankyrase function and will also reveal new roles. While we focused on the 74 described above, our screen yielded a total of 608 proteins with a significant change in abundance; some proteins increased, while a similar number decreased. A number of proteins that show a significant increase lack a RxxG(P/A/C)xG tankyrase binding site. These proteins may contain a cryptic motif. Alternatively, they could bind tankyrase through novel interaction modes or interact indirectly. In addition, proteins could accumulate by mechanisms that are distinct from PAR-dependent ubiquitylation. Regarding proteins that decreased in abundance, some (such as Nestin) may result indirectly from proteins stabilized in the absence of

tankyrase. Bioinformatic analysis for protein-protein interactions among the 608 proteins will likely identify potential connections that can be experimentally validated.

Considering the limited number of known targetable enzymes in Wnt/β-catenin signaling, the identification of tankyrase as a druggable node in the pathway opened up new possibilities[20, 56]. Here we identified two tankyrase targets in the Wnt/β-catenin signaling pathway HectD1 and NKD2 (and NKD1). Both (like Axin) are negative regulators of the pathway. HectD1, modifies APC with K63-ubiquitin to stabilize the APC-Axin interaction and promote degradation of β-catenin[39]. NKD2 promotes degradation of DVL-1 to prevent accumulation of β-catenin[45]. Tankyrase-mediated degradation of HectD1 or NKD2 would have a similar impact on cancer cell growth as degradation of Axin, to stabilize β-catenin and promote growth. Thus, inhibition of tankyrases could influence β-catenin levels through multiple arms of the pathway, perhaps explaining why the reduction of β-catenin is so robust in the DKO cells. Tankyrase small-molecule inhibitors may have an advantage, by targeting multiple arms of the pathway. Recent studies suggest that NKD may be a viable target. NKD2 expression was found to suppress tumor growth and to be downregulated in human metastatic osteosarcoma cells[57]. One recent study found that the tankyrase-specific inhibitor JW74 induced apoptosis in osteosarcoma cell lines[58], we speculate that it could be through stabilization of NKD2.

Unexpectedly, we identified the Notch signaling pathway in our screen. Notch is a single-pass transmembrane receptor. Upon activation by ligand, the extracellular domain is cleaved, followed by γ-secretase cleavage in the membrane to release the Notch-ICD, which enters the nucleus to release repression and activate transcription of target genes. We showed by immunoblot analysis that the intracellular portion of Notch2 was stabilized in tankyrase knockout cells. Immunofluorescence analysis showed increased Notch2 localized to the plasma membrane, suggesting that it was the uncleaved membrane-bound form of the intracellular portion that was stabilized in the absence of tankyrases. This pattern of increased Notch2 staining on the plasma membrane was obtained by treating cells with a tankyrase-specific inhibitor and was also observed upon treatment with γ-secretase inhibitor. Immunoblot analysis showed that activated Notch2-ICD, which is the form that goes to the nucleus to activate transcription, was decreased upon inhibition of tankyrases or γ-secretase. Moreover, we observed a decrease in mRNA levels of the Notch target Nestin in tankyrase double knockout cells that was recapitulated in cells treated with tankyrase or γ-secretase inhibitors. Together our studies indicate that release of the Notch2-ICD from the plasma membrane and subsequent localization and function in the nucleus depends upon tankyrases. Considering that the Notch signaling pathway (like the Wnt/β-catenin pathway) is commonly activated in cancer[59], tankyrase inhibitors may have therapeutic potential in targeting both pathways.

## Methods

**Generation of human tankyrase knockout cell lines**. The RNA-guided CRISPR associated nuclease Cas9 was used to introduce targeted loss-of-function mutations in specific sites in the *TNKS* genes[60, 61]. For *TNKS1* KO, a 20 bp target sequence directed against the first exon of the human *TNKS1* gene (*TNKS1* Guide DNA 5′-CGATCCCCGGACCCGGTTGA-3′) was inserted into the guide sequence insertion site of the CRISPR plasmid pX330 comprised of Cas9 and a chimeric guide RNA and used to transfect HEK293T cells. Following transfection, cells were re-plated for single-cell cloning, propagated, and screened by a PCR strategy designed to screen for loss of a HincII site in the target site. Twenty-four lines were screened: 7 showed HincII resistance and 3 of the 7 showed complete loss of tankyrase 1 protein by immunoblot. DNA sequencing of the PCR products from clones #3, #13, and #15 confirmed insertion or deletions leading to stop codons. *TNKS1* KO clone #13 was used for the analyses in Figs. 1 and 6.

For *TNKS2* KO and *TNKS1* KO/*TNKS2* KO a 20 bp target sequence directed against the first exon of the human *TNKS2* gene (*TNKS2* Guide DNA 5′-CTGTTCGAGGCGTGCCGCAA-3′) was inserted into the pX330 CRISPR plasmid and used to transfect HEK293T cells to generate *TNKS2* KO cells and used to transfect *TNKS1* KO (#3) cells to generate *TNKS1* KO/*TNKS2* KO cells. Following transfection, cells were re-plated for single-cell cloning, propagated, and screened by DNA sequencing to identify insertions/deletions leading to stop codons. Loss of tankyrase 2 protein was confirmed by immunoblot analysis.

**Preparation of extracts for the proteomic screen**. Three 15 cm dishes each of HEK293T WT and DKO cells at 70–80% confluency were washed with ice cold 1× phosphate-buffered saline (PBS) and harvested using lysis buffer containing 50 mM HEPES (pH 8.5), 25 mg/ml NaDOC, 8 M Urea, 2.5% protease inhibitor cocktail (PIC) (Sigma) and phosphatase inhibitor cocktail 3 (Sigma).

**Protein digestion**. The protein lysates were measured at 280 nm using the NanoDrop DS-11 Spectrophotometer (DeNovix) to determine the protein concentration. To reduce the disulfide bonds, we added reducing agent (5 mM dithiothreitol (DTT)) for 1 h at 55 °C. The cysteines were subsequently alkylated by a 45 min incubation in the dark with iodoacetamide (14 mM) at 55 °C. The reaction was quenched using an additional aliquot of reducing reagent. The protein lysate was first incubated at pH 8.5 with Lys-C (Promega) at a 200:1 (protein:enzyme) ratio for 120 min at room temperature in 8 M urea and 10 mM Tris-HCL. Next, 10 mM Tris-HCl (pH 8.5) were added to dilute the urea concentration to 2 M. After dilution the protein lysate was digested with Trypsin (Promega) at a 100:1 (protein:enzyme) ratio for 8 h. The pH of the digested protein lysate was lowered to pH < 3 using trifluoroacetic acid (TFA). The digested lysate was desalted using C18 solid-phase extraction (Sep-Pak, Waters). 80% acetonitrile (ACN) in 0.5% acetic acid was used to elute the desalted peptides. The peptide eluate was concentrated in the Speedvac and stored at −80 °C until further analysis.

**TMT labeling**. The dried peptide mixture was re-suspended in 50 mM HEPES (pH 8.5) using a volume of 70 µl. From each sample aliquots of 180 µg were labeled with TMT reagent according to the manufacturer's protocol. In brief, each TMT reagent vial (0.8 mg) was dissolved in 44 µL of ACN and 10 µL was added to each sample with the addition of 20 µL of ACN. The reaction was allowed to proceed for 60 min at room temperature and then quenched using 4 µL of 5% w/v hydroxylamine. The samples were combined at a 1:1 ratio and the pooled sample desalted over SCX and SAX solid-phase extraction columns (Strata, Phenomenex).

**Off-line basic-pH RP fractionation**. The pooled sample was fractionated using basic pH reverse-phase HPLC (Buffer A = 10 mM ammonium formate, pH 10.0; Buffer B = 90% ACN, 10 mM ammonium formate, pH 10.0) on a 4.6 mm × 250 mm Xbridge C18 column (Waters, 3.5 µm bead size) using an Agilent 1260 Infinity Bio-inert HPLC. The peptide mixture was separated over a 60 min linear gradient from 10 to 50% solvent B at a flow rate of 0.5 ml/min. A total of 90 fractions were collected. Combining equal volumes of early, middle and late eluting fractions the 90 fractions were concatenated into 30 final fractions. The fractions were concentrated in the Speedvac and stored at 80 °C until further analysis.

**LC-MS/MS analysis**. An aliquot of each concatenated fraction was loaded onto a trap column (Acclaim® PepMap 100 pre-column, 75 µm × 2 cm, C18, 3 µm, 100 Å, Thermo Scientific) equilibrated with solvent A (2 % acetonitrile, 0.5% acetic acid) connected to an analytical column (EASY-Spray column, 50 m × 75 µm ID, PepMap RSLC C18, 2 µm, 100 Å, Thermo Scientific) using the autosampler of an Easy nLC 1000 (Thermo Scientific). After equilibrating the sample with 5% solvent B (90% acetonitrile, 0.5% acetic acid), the sample was eluted into the Q Exactive mass spectrometer (Thermo Scientific) using a 120 min linear gradient from 5%-30% solvent B with a flow rate of 200 nl/min. The full scan was recorded with a resolution of 70,000 (@ *m/z* 200), a target value of 1e6 and a maximum ion time of 50 ms. After each full scan 20 MS/MS scans were recorded on the top 20 ions using the following parameters: resolution 35,000 (@*m/z* 200), isolation window of 1.6 *m/z*, normalized collision energy of 27, HCD fragmentation, target value of 5e4, maximum ion time of 180 ms, enabled monoisotopic precursor selection and dynamic exclusion of 30 s.

**Data analysis**. The mass spectrometry raw data were processed using MaxQuant[62] version 1.5.2.8. Proteins and peptides were searched against the UniProt human FASTA database using a target-decoy approach with the Andromeda[63] search engine integrated into the MaxQuant environment using the following settings: oxidized methionine (M), TMT-labeled N-term and lysine, acetylation (protein N-term) and deamidation (asparagine and glutamine) were selected as variable modifications, and carbamidomethyl (C) as fixed modifications; precursor mass tolerance was set to 10 ppm; fragment mass tolerance was set to 0.01 Th; The identifications were filtered using a false-discovery rate (FDR) of 0.01 at the level of proteins and peptides, and minimum peptide length of six amino acids. For quantification the following criteria and filters were used: (1) only unique peptides were used for quantification and only proteins with at least two unique peptides

were reported. (2) A minimum reporter ion intensity of 1000 was required. Data analysis was performed using Perseus, Microsoft Excel and R statistical computing software. Each reporter ion channel was summed across all quantified proteins and normalized assuming equal protein loading across all 10 samples. To identify proteins that are differentially expressed between the wild-type and double knock out mutant a Welch's $t$-test was applied followed by a permutation-based FDR to control for multiple testing error.

**Plasmids.** TNKS lentiviral plasmids contain full-length Flag-epitope tagged TNKS1 or TNKS2 cloned into lentiviral vector pLKO.1ps[64, 65]. The TRF1 plasmid contains full-length myc-epitope tagged TRF1 cloned into retroviral vector pLPC[66]. GFP-NKD2 contains N-terminal GFP fused to NKD2 cloned into pEGFP-N2 vector (provided by Robert Coffey)[67]. The GFP-NKD2 mutation was created by substituting the glycine (G) at position 21, with arginine (R) by site-directed mutagenesis of GFP-NKD2 using the oligonucleotide 5'-CGGAGAGAGAGCCC GGAAAGGGACAGCTTCGTGGCG-3'. HA-NKD1 contains N-terminal HA epitope-tagged NKD1 cloned into pCDNA3.1 vector (provided by Wanguo Liu). Flag-Notch2 contains N-terminal Flag-tagged notch intracellular domain (ICD) cloned into p3XFLAG-CMV-7 vector (provided by Raphael Kopan) (Addgene plasmid # 20184)[68]. The Flag-Notch2 mutation was created by substituting the glycine (G) at position 1731, with arginine (R) by site-directed mutagenesis of Flag-Notch2 using the oligonucleotide 5'-CGCCGTGAACCTGTGCGACAGGATGC CGTGG-3'. Mutagenesis was performed using the Stratagene QuikChange site-directed mutagenesis kit according to the manufacturer's instructions.

**Cell lines.** HEK293T (ATCC), HEPG2 (ATCC), HeLa.I.2.11[69], and MCF7 (ATCC) cells were grown under standard conditions. Where indicated the following inhibitors were added: #8 and #15 (MolPort) (provided by Lari Lehtio)[44] at 10 μM final concentration for 16 h, and XAV939 (SelleckChem) (under low serum conditions), and DAPT (Sigma) at 10 μM final concentration for 16 h.

**Lentiviral infection.** Rescue cell lines were generated by introducing lentiviruses expressing vector (V), TNKS1 (T1), TNKS2 (T2), or both (T1T2) into DKO cells at PD 120. For lentivirus generation, 293FT cells (Invitrogen) were transfected using Lipofectamine 2000 (Invitrogen) with 1 μg each lentiviral vector and pCMVΔR.89 packaging plasmid, and 100 ng pMD.G envelope plasmid. Forty-eight hours after transfection, supernatants were collected, filtered with a 0.45-μm filter (Millipore), supplemented with 8 μg/ml polybrene (Sigma-Aldrich), and used to infect target cells. Following 48–72 h infection, cells were sub-cultured 1:2 into medium containing 2 μg/ml puromycin.

**Plasmid and siRNA transfection.** For plasmid transfection, cells were transfected with Lipofectamine 2000 (Invitrogen) according to the manufacturer's protocol for 18–20 h. For siRNA transfection cells were transfected with RNF146–2 (5'-GGAUGUAUCUGCAGUUGUU-3')[21] or GFP Duplex I (Dharmacon Research Inc.) (final concentration of 100 nM) with Oligofectamine (Invitrogen) according to the manufacturer's protocol for 48 h.

**qRT-PCR.** Total RNA was isolated from HEK293 WT, *TNKS1* KO, *TNKS2* KO, and DKO cells using Trizol reagent (Invitrogen) according to the manufacturer's instructions. Aliquots of 1 μg RNA were used for reverse transcription with random priming (Protoscript First Strand cDNA Synthesis Kit, NEB) as per the manufacturer's instruction. Real-time PCR reactions were set up with LightCycler 480 SYBR Green 1 master (Roche) using 2% of each cDNA preparation. Relative hTERT expression levels were obtained by normalizing to Pumilio expression levels. Primer pairs were *Pumilio* (5'-CGGTCGTCCTGAGGATAAAA-3'; 5'-CGT ACGTGAGGCGTGAGTAA-3') and hTERT (5'-CGGAAGAGTGTCTGGAGCA A-3'; 5'-GGATGAAGCGGAGTCTGGA-3'). Relative Nestin levels were obtained by normalizing to GAPDH levels. Primer pairs were GAPDH (5'-AGCCACA TCGCTCAGACAC-3'; 5'-GCCCAATACGACCAAATCC-3') and Nestin (5'-GGG AAGAGGTGATGGAACCA-3'; 5'-AAGCCCTGAACCCTCTTTGC-3')[70].

**Cell extracts.** Four volumes of TNE buffer (10 mM Tris (pH 7.8), 1% Nonidet P-40, 0.15 M NaCl, 1 mM EDTA, and 2.5% protease inhibitor cocktail (PIC) (Sigma)) was used to resuspend cell pellets. Following a 1 h incubation on ice, suspensions were pelleted at 8000 × g for 15 min. Supernatants were fractionated by SDS–PAGE and analyzed by immunoblotting.

**Immunoblot analysis.** Immunoblots were incubated separately with the following primary antibodies: rabbit anti–tankyrase1 762 (1 μg/ml)[16]; tankyrase1 465 (4 μg/ml)[9], rabbit anti-Flag (1 μg/ml) (Sigma Aldrich, F7425); rabbit anti-c-Myc (0.2 μg/ml) (Santa Cruz, sc-789); mouse anti-α-tubulin ascites (1:10,000) (Sigma Aldrich, T5768); rabbit anti-Hectd1 (1:1000) (Bethyl, A302-908A-T); rabbit anti-Naked1 (C30F10) (1:1000) (Cell Signaling, 2201); rabbit anti-HA (0.5 μg/ml) (Abcam, ab9110); rabbit anti-Naked2 (C67C4) (1:1000) (Cell Signaling, 2073); rabbit anti-VAMP8 (1:1000) (Bethyl A304-350A-T)or Abcam 76021 (1:1000); rabbit anti-DICER (1:1000) (Cell Signaling, 3363); rabbit anti-Notch2 (C-terminal) (D76A6) XP (1:1000) (Cell Signaling, 5732); rabbit anti-Notch2 (cleaved Val1697) (1:1000)

(Sigma SAB4502022); rabbit anti-Notch1 (D6F11) (1:1000) (Cell Signaling 4380); rabbit anti-Notch3 (D11B8) (1:1000) (Cell Signaling 5276); rabbit anti-RNF146 (2.0 μg/ml)(Abcam ab106334); rabbit anti-Angiomotin (1:5000) (Bethyl, A303-305A-T-1); rabbit anti-Axin1 (C95H11) (1:1000) (Cell Signaling, 2074); mouse anti-HP1Δ (1:5000) (Millipore, MAB3450); rabbit anti-Chk2 (H-300) (0.2 μg/ml) (Santa Cruz, sc9064); or rabbit anti-GFP (2 μg/ml) (Abcam, ab290), followed by horseradish peroxidase-conjugated donkey anti-rabbit or anti-mouse IgG (Amersham) (1:2500). Bound antibody was detected with Super Signal West Pico (Thermo Scientific). Uncropped scans of immunoblots appear in Supplementary Figure 4.

**Immunoprecipitation.** Cells were lysed as above and supernatants precleared with Protein G-Sepharose rotating at 4 °C for 30 min. Nonspecific protein aggregates were removed by centrifugation and the supernatant was used for immunoprecipitation analysis or fractionated directly on SDS-PAGE (indicated as input, ~5% of the amount used in the immunoprecipitation). For immunoprecipitation of Myc, Flag, or HA epitope-tagged proteins, supernatants were incubated with 20 μl of rabbit-Anti-c-Myc Agarose Affinity Gel antibody (Sigma, A7470) or Flag M2 agarose (Sigma, A2220) for 3 h. For all other immunoprecipitations supernatants were incubated for 3 h with 1 μg rabbit anti-GFP (Abcam, ab290), Rabbit anti-tankyrase 465[9], rabbit anti-HA (Abcam, ab9110), or IgG, followed by Protein G-Sepharose for 1 h. For all immunoprecipitations, beads were washed three times with 1 ml of TNE buffer, fractionated by SDS-PAGE, and processed for immunoblotting as described above.

**Indirect immunofluorescence.** Following fixation in 2% paraformaldehyde in PBS and permeabiization in 0.5% NP40 in PBS for 10 min each, cells on coverslips were blocked in 1% BSA in PBS, followed by incubation with rabbit anti-Notch2 (C-terminal) (D76A6)XP (1:100) (Cell Signaling, 5732), rabbit anti-Notch3 (D11B8) (1:100) (Cell Signaling 5276), or mouse anti-α-tubulin ascites (1:5000) (Sigma Aldrich, T5768) antibodies. Fluorescein isothiocyanate- conjugated donkey anti-rabbit or anti-mouse antibodies (1:100) (Jackson Laboratories) were used to detect primary antibodies. DNA was stained with DAPI (0.2 mg/ml).

**Chromosome-specific FISH.** Methanol:acetic acid (3:1) was used to fix cells for 15 min twice[27]. Cells were collected on slides in a cytospin (Shandon) (2000 rpm for 2 min), rehydrated at 37°C for 2 min in 2× SSC, and dehydrated in an ethanol series of 70%, 80% and 95% for 2 min each. Following denaturation at 75°C for 2 min, cells were hybridized overnight at 37 °C with a 16p subtelomeric FITC-conjugated probe (Cytocell). Cells were washed at 72 °C for 2 min in 0.4× SSC and at RT for 30 s in 2X SSC with 0.05% Tween 20. DNA was stained with 0.2 μg/ml DAPI. Telomeric loci in mitotic cells were considered cohered if 50% or more appeared as singlets, for example one out of two or two out of three.

**Telomere restriction fragment analysis.** Genomic DNA was isolated from HEK293T TNKS KO cell lines and digested with *Hinf*I, *Alu*1, *Mbo*I, and *Rsa*I. Approximately 3 μg of the digested DNA was fractionated on 1% agarose gels using pulsed-field gel electrophoresis. Telomeres were detected by hybridization to a $^{32}$P end-labeled (CCCTAA)$_3$ oligonucleotide probe as described[27]. The mean telomere length was determined using Telometric (Fox Chase Cancer Center).

**PARP assays.** Immunocomplexes bound to beads were washed three times with HBS buffer (20 mM Tris, pH 5.5, 0.5% Nonidet P40, 0.5 M NaCl and protease inhibitor cocktail), washed once 1 mM 3AB (3-aminobenzamide), washed once in 100 μl of PARP reaction buffer (50 mM Tris, pH 8.0, 4 mM MgCl$_2$, 0.2 mM dithiothreitol), and incubated in 100 μl of PARP reaction buffer containing 50 μCi of [32 P]NAD$^+$ (1000 Ci/mmol; Perkin Elmer) for 30 min at 25 °C without or with 0.2 μg of recombinant tankyrase 1 (Trevigen). Reactions were terminated by the addition sample buffer and fractionated by SDS/PAGE. Gels were stained with Coommasie blue and autoradiographed.

**Image acquisition.** Microscopy was performed with an Axioplan 2 microscope fitted with a Plan Apochrome 63x NA 1.4 oil immersion lens (Carl Zeiss, Inc.) and a C4742-95 digital camera (Hamamatsu Photonics). Openlab software (Perkin Elmer) was used to acquire and process images. Multiple planes were merged if chromosome-specific FISH foci fell in more than one optical plane of focus.

**Statistical analysis.** Student's unpaired $t$-test was applied using Prism 6 software. Data are given as mean ± SD (standard deviation) or as mean ± SEM (standard error of the mean); $P < 0.05$ values were deemed significant.

**Data availability.** The mass spectrometry raw files are accessible under MassIVE ID: MSV000081663 and ProteomeXchange ID: PXD008117. The authors declare that all data supporting the findings of this study are available with the article and its supplementary information or from the corresponding author upon request.

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

## Acknowledgements

We thank the Smith Lab for critical reading of the manuscript, Robert Coffey and Wanguo Liu for providing plasmids, and Lari Lehtio for tankyrase inhibitors. The mass spectrometric experiments and data analysis were supported in part by NYU School of Medicine, the Laura and Isaac Perlmutter Cancer Center Support grant P30CA016087 from the National Cancer Institute and a shared instrumentation grant from the NIH, 1S10OD010582-01A1 for the purchase of an Orbitrap Fusion Lumos. Research reported in this publication was supported by the NCI of the NIH under award number R01CA116352 to S.S.

## Author contributions

A.B., Y.Y., B.U., and S.S. designed experiments and analyzed the data. A.B. and Y.Y. performed experiments. S.S. wrote the manuscript. A.B. and B.U. read and corrected the manuscript. S.S. provided the research funds. All authors approved the final manuscript.

## Additional information

**Competing interests:** The authors declare no competing financial interests.

