## [Peer Review File · Nature Communications]

Reviewers' comments:

Reviewer #1 (Remarks to the Author):

Tankyrases are involved in a wide range of biological functions which also include potential therapeutically targetable pathways and processes. However, we do not fully appreciate the complexity of pathways controlled by the tankyrases. Bhardwaj et al. report the generation of TNKS, TNKS2 and TNKS/TNKS2 knockout HEK293T cell lines by CRISPR. By performing a proteome-wide abundance study, they make significant headway into addressing this important open question.

To start with, the authors show that both tankyrases are required for sister telomere resolution in mitosis but that they can compensate for one another in their role to maintain telomere length. Moreover, the authors confirm a role of TNKS in spindle formation and demonstrate that TNKS2 is also required for normal spindle assembly. This confirms and further establishes an involvement of tankyrases in these processes.

By performing a quantitative whole-proteome study, the authors identify around 300 proteins that accumulate and another around 300 proteins that decrease in abundance upon loss of both tankyrases. By focusing on accumulating candidates containing putative tankyrase-binding motifs, the authors identify a range of known tankyrase targets but also a large number of novel potentially tankyrase-regulated proteins, including components of the Wnt/beta-catenin and Notch signalling pathways. A selection of these are then validated as interactors, with a focus on NKD2, NKD1 and NOTCH2. Interestingly, loss of TNKS (but not TNKS2) gives rise to an accumulation of NOTCH2 at the plasma membrane and decreased Notch signalling, as measured by target gene transcription, pointing to important roles of TNKS in this therapeutically interesting pathway and providing one of the few examples where the two tankyrases may have distinct functions.

This work is novel, timely and makes an important, long-awaited contribution to the field. The proteomics findings and TNKS/TNKS2 knockout cells will be an excellent resource. This impactful work will attract the attention from a wide readership studying the diverse functions of tankyrase.

Specific points:

(1) When discussing their proteomics work, the authors focus on a selection of tankyrase-regulated proteins, namely those containing potential tankyrase-binding motifs. A certain motif degeneracy means that these motifs can relatively easily be missed, as illustrated by the literature. (On that note, how were tankyrase-binding motifs identified? Were the candidate targets screened bioinformatically?) Moreover, there may be tankyrase targets that lack the known tankyrase-binding motif and interact with tankyrase either indirectly or via novel interaction modes, in addition to proteins that accumulate via mechanisms distinct from PAR-dependent ubiquitination. This would also make an important point for the Discussion section.

From the selected subset of proteins, the full complement of tankyrase-regulated processes cannot be fully appreciated. While Table S1 lists all identified 7254 proteins, it would be valuable to extend the discussion to those proteins whose abundance changes substantially but that do not have a clear tankyrase-binding motif. (Such motifs could be either absent or cryptic.) The authors could also point out those targets with the largest fold-changes in abundance in Figure 2D (around 2 dozens). It would certainly be appreciated to make the supplemental tables as accessible as possible and provide a table legend defining the various columns. The columns "Gene Name" and "Protein Name", as included in Table S2, are very helpful, and could be added to Table S1 as well. Upon surveying those proteins that accumulate upon loss of both TNKS and TNKS2, what is the estimated percentage of proteins that bear a detectable tankyrase-binding motif?

(2) An update for the Introduction (page 3): olaparib has been approved by the FDA and EMA for

treatment of germline BRCA-mutated advanced ovarian cancer, in addition to ongoing clinical trials. Likewise, rucaparib has been FDA-approved.

(3) Minor point, Introduction (page 3): the correct name of the SAM domain is "sterile alpha motif domain".

(4) Introduction (page 3) and subsequent manuscript: the tankyrase-binding motif extends to a maximum of eight (rather than the minimal six) amino acids. While a rigid consensus probably "misses" degenerate motifs, reported consensus sequences, derived from experimental approaches and sequence analysis, include Arg-x-x-[small hydrophobic amino acids / Gly]-[Asp/Glu, in addition to a small selection of other tolerated amino acids]-Gly-[no Pro]-[Asp/Glu] (Guettler et al., 2011). While not every motif is eight amino acids long, it appears that the eighth position may be relevant to some of the novel tankyrase target candidates reported here, such as NOTCH1-4, which feature an aspartate residue at position eight, which almost certainly will contribute to binding. In Figure 2E, can the motif sequence shown be extended accordingly?

(5) Page 5, first paragraph: can you mention all KOs you generated in the first sentence?

(6) Page 5, second paragraph and Discussion: an alternative interpretation of a requirement for both tankyrases is total tankyrase level (dosage). Where knockout of a single tankyrase leads to a phenotype (telomere cohesion, Notch localisation), it remains possible that these processes require a higher overall level of tankyrase, regardless of any potential differences between TNKS and TNKS2. In this case, the level of either tankyrase protein alone may not be sufficient. Differential phenotypes for single knockouts may simply reflect different expression levels of TNKS vs. TNKS2. In other words, losing the more abundant tankyrase protein will result in a more striking phenotype.

(7) Figure 2A: the effect on beta-catenin levels does not seem particularly striking (compare tubulin loading control and beta-catenin panel), although it is detectable. Can this blot and replicate blots be quantified? From the Methods section, it appears that total cell lysates were analysed, which would include membrane-associated beta-catenin. This may explain the subtle effect, given that the bulk of cellular beta-catenin is membrane-associated and not subject to regulation by Wnt/beta-catenin signalling. The authors may point out in the figure legend that total lysates were analysed.

(8) Page 8, paragraph 2: the term "in vivo" may not be appropriate for experiments in cell lines. Simply stating that the interaction of endogenous proteins was analysed would be sufficient.

(9) Page 11, Discussion, paragraph 1: "heteropolymer" may be the more appropriate description, rather than "heterodimer", since tankyrases can self-associate into heterotypic oligomers (De Rycker et al., 2004; Mariotti et al., 2016; Riccio et al., 2016).

(10) Minor point (typo), page 11, Discussion, paragraph 2: "grown" instead of "gown"

(11) Page 12, Discussion, paragraph 1: the actual reason for why Wnt/beta-catenin signalling is challenging to target may be its complexity and the redundancy of components, its wide involvement in adult tissue stem cell maintenance (giving rise to toxicity of pathway inhibitors) and the limited number of targetable components, rather than the fact that it is highly mutated in cancer.

In the same paragraph, the discussion that NKD2 induction may underlie the effect of JW74 on apoptosis is perhaps a bit too speculative. It might be removed or discussed as speculation, unless other data are available, but at least the cited paper does not appear to include such data.

(12) Minor point, figure 1, legend: HEK293T instead of Hek293T

(13) Figure 2, legend: the term "reduction" rather than "loss" of beta-catenin appears more appropriate. (See comments above on labelling panel D and displaying an extended Tankyrase-binding motif in panel E of figure 2.)

(14) Figure 3D, legend: the term "plot" rather than "graphical representation" may be more appropriate.

(15) Figure 5A, legend: can you define the abbreviation EFX?

(16) Figure 6, legend: can you define the abbreviations RAM and RBPJ (here and in the text) and correct a typo (should read: "...tankyrase binding site across human Notch 1-4 ...", delete "in"). Also, the legend currently misses an explanation of the asterisks in panel E. In panel E, can the heavy and light IgG chains be indicated and a typo (correct spelling "Coomassie") be addressed? The figure states that TRF1 is Myc-tagged while the legend refers to Flag-TRF1.

(17) Minor point: would the term "PARylation" be preferable to "PARsylation"? I.e., where does the "s" come from?

Reviewer #2 (Remarks to the Author):

To the authors:

The ms by Bhardwaj et al addresses novel target proteins for tankyrases, and also addresses the specific roles of tankyrase 1 and 2 via CRISPR/Cas9 targeting. They provide data on the specific roles of tankyrase 1 and 2 for telomere length, telomere cohesion and mitotic spindle integrity. They also conduct a proteomics analysis, using isomeric tag-based t-mass, which reveals novel target proteins, including the Notch 2 receptor. This prompted the authors do make a more deep-drilling effort into understanding the link between tankyrase and Notch (Notch2), and they provide evidence for direct interaction and that tankyrase ADP-ribosylates Notch2. Furthermore, the routing of Notch2 from the plasma membrane to the nucleus is affected by the tankyrase status. This represents a novel mode of posttranslational modification of the Notch intracellular domain, and may be important for future targeting of the Notch signaling pathway, or at least to understand possible adverse side-effect of tankyrase inhibitors.

Specific comments:

I think that the authors convincingly show that Notch2 is regulated by tankyrase (including mutating the putative binding site in Notch2). They also demonstrate that at least Notch1 and 3 are putative targets for tankyrase. It would be valuable to see whether these two Notch receptors also are de facto regulated by tankyrase. One reason for emphasizing on this experiment is that recent work on aPKC and PIM kinases have shown that they posttranslationally modify only specific Notch receptor paralogs, and potential differences in response to tankyrase among N1-N3 would aid in understanding possible paralog-specific functions.

Figure 3A. How strong is the increase in Notch2 following tankyrase inhibitor exposure? Is it significant? The tubulin band is considerably stronger in the WT+Ti15 lane, and it would be good to see a densitometric scan + ratio to tubulin across all three lanes

Figure 3B. Why was not an inhibitor experiment (similar to Figure 3A) included also for Dicer, Chk2 and VAMP8?

Figure 3C. The change of cell line in Figure 3C (to HepG2 cells) and the use of different inhibitors only tested on Dicer is not explained in the text. Figure 3A-C thus gives a patchy impression, and it would be good to see that the effects are not restricted to specific cell lines/genes/inhibitors.

Figure 6B. The authors argue that there is more Notch2 ICD. By using the Cell signaling antibody it is not possible to distinguish Notch2 ICD from a membrane-tethered (NEXT) form of Notch2. There are now antibodies that identify the "neo-epitope" appearing in Notch2 ICD after S3 cleavage, and

it would be important to use such an antibody to back up this claim.

Figure 6F. The authors demonstrate a strong upregulation of plasma membrane-associated Notch2 staining in the tankyrase-deficient cells, which is interesting. As Notch receptors do not exclusively localize to the plasma membrane or nucleus, but a fair amount also resides in endocytic vesicles, it would be good to see if tankyrase also affects the localization to endosomes. This could be achieved by double-staining with markers for endosomes. To more precisely measure the amount of Notch2 receptor at the cell surface in response to tankyrase, a suggestion would be to use cell surface biotinylation for the Notch2 receptor. This would give a precise estimate of the extent of cell surface localization.

Figure 6F. Along the same lines, what would happen if a Notch2 construct similar to Notch1 Δ E, which is a truncated transmembrane form which does not require ligand activation but needs g-secretase processing, is used? Would such a construct also accumulate at the cell surface?

An intriguing idea is whether tankyrase is "preloaded" on the intracellular domain of the Notch receptor already prior to cleavage of the Notch ICD. This could be tested by addressing whether tankyrase binds to Notch also when Notch S3-processing is blocked (by using g-secretase inhibitors such as DAPT).

Figure 6H. The choice of nestin as a downstream Notch target gene is unconventional. There is indeed a report that nestin can be regulated by Notch in certain contexts, but there are a number of more established downstream genes, such as Hes or Hey genes, or NRARP. It would be good to see the nestin data complemented by these genes as well. In addition, the use of a direct downstream reporter construct, such as a multimerized CSL-binding site linked to luciferase, would provide a complementary type of analysis of downstream activation.

Minor comments:

Discussion: The discussion on lethality in vivo for the double tankyrase OK embryos versus survival of the HEK293 cells in which T1 and T2 are removed seems a bit naïve – it really takes a lot to kill an established cell line such as HEK293.

Mice lacking both T1 and T2 are embryonically lethal – is there by any chance phenotypes which resemble increased Notch signaling. A good experiment would be to look for higher levels of Notch2 in the double knockout mouse embryos (although I realize that it is a daunting experiment if the authors do not have access to the mice, or can obtain embryo material from another group).

The authors cite the reviews by Carrieri and Dale 2016 and Nowell and Radtke 2017 for Notch oncogene reviews. The Nowell/Radtke review basically addresses the tumor suppressor function and a more recent, and general, review would be Aster, Pear, Blacklow Ann. Rev Pathol 2017.

Reviewer #3 (Remarks to the Author):

In the manuscript entitled "Whole proteome analysis of human tankyrase knockout cells reveals new targets of tankyrase-mediated degradation", the authors established a TNKS1/2 double knockout cells in HEK293T cell line by using CRISPR/Cas9, and systematically analyzed the degrading substrates for TNKS1/2 via TMT-based proteomics. Through it, they not only validated previously identified TNKS1/2 substrates for degradation, but also uncovered novel candidate substrates in different pathways or cellular functions (WNT, miRNA, and NOTCH). Overall, this study is very interesting and provides a useful resource for the field to further characterize the cellular functions of TNKS1/2.

Major points:

1. In Fig 1, the authors claimed that they generated TNKS1/2 single and double KO cells, and functionally validated these cells. In addition, they took advantage of these established KO cells to compare the roles between TNKS1 and TNKS2 by using telomere and mitosis-related assays. It

seems that TNKS1 and 2 showed redundant functions in all these assays (Fig1C, 1F and 1H). If the authors really want to claim that TNKS1 and TNKS2 are both required for some cellular functions (i.e. telomere cohesion and mitotic spindle integrity), they at least need to 1) perform the significance analysis (p value) between DKO cell and single KO cells in Fig. 1C and Fig.1H; and 2) show rescue data in DKO cells by using TNKS1 or TNKS2 alone or in a combination.

2. As for the TMT-MS validation data in Fig3, the authors used tankyrase inhibitors (compounds 8 and 15) as control to validate the candidate expression in DKO cells. Since tankyrase inhibitors target tankyrase-enzyme activity to stabilize the substrates, the DKO should have at least the close or even higher effect than the inhibitors to stabilize the substrates. In that case, why did AMOT and NKD2 have higher expressions in inhibitor-treated condition than in the DKO cells? As for Fig3A, inhibitor treatment should also be performed in DKO cells to serve as a control.

3. In Fig4A, the authors performed rescue experiments in DKO cells, which is quite important and helpful to validate their DKO cells. Since tankyrase PAR activity is required to target substrate for degradation, it will be great if a PAR-deficient mutant of TNKS could be included here as a control.

4. In Fig6, the authors validated Notch receptors as targets for TNKS1/2-mediated degradation. The biochemical experiments are fine, but I am confused about the notch-related functional studies presented here. Based on the provided data, it seems that TNKS1/2 can not only degrade Notch2 but also activate Notch signaling by enhancing Notch2 nuclear localization. This discrepancy needs clarification.

Compared to DKO and Ti8-treated cells, Notch2 seems to be undetected in WT cells. Based on the figures provided (Fig6F and 6G), I could see more nuclear staining of Notch2 in DKO and Ti8-treated cells compared to that in WT cells.

As for Fig6H, there is no direct evidence provided here to show that downregulation of Nestin transcription is through TNKS1/2-mediated alteration of Notch2 (degradation? membrane retention?).

Minor points:

1. In Fig1G and 1H, if >40% DKO cells have mitotic defects, how would the authors establish and obtain the DKO cells? Is there any cell proliferation difference between WT, single and double KO cells?

2. In Fig2A, the tubulin blot seems also decreased in KO and DKO cells.

3. In Fig6C, what is "tx"? I guess that it is short for "transfected".

4. In Fig6G, can γ -seretase inhibitor stabilize Notch2? Its treatment seems to have the similar effect on Notch2 protein level to Ti8.

Reviewers' comments:

Reviewer #1 (Remarks to the Author):

Tankyrases are involved in a wide range of biological functions which also include potential therapeutically targetable pathways and processes. However, we do not fully appreciate the complexity of pathways controlled by the tankyrases. Bhardwaj et al. report the generation of TNKS, TNKS2 and TNKS/TNKS2 knockout HEK293T cell lines by CRISPR. By performing a proteome-wide abundance study, they make significant headway into addressing this important open question.

To start with, the authors show that both tankyrases are required for sister telomere resolution in mitosis but that they can compensate for one another in their role to maintain telomere length. Moreover, the authors confirm a role of TNKS in spindle formation and demonstrate that TNKS2 is also required for normal spindle assembly. This confirms and further establishes an involvement of tankyrases in these processes.

By performing a quantitative whole-proteome study, the authors identify around 300 proteins that accumulate and another around 300 proteins that decrease in abundance upon loss of both tankyrases. By focusing on accumulating candidates containing putative tankyrase-binding motifs, the authors identify a range of known tankyrase targets but also a large number of novel potentially tankyrase-regulated proteins, including components of the Wnt/beta-catenin and Notch signalling pathways. A selection of these are then validated as interactors, with a focus on NKD2, NKD1 and NOTCH2. Interestingly, loss of TNKS (but not TNKS2) gives rise to an accumulation of NOTCH2 at the plasma membrane and decreased Notch signalling, as measured by target gene transcription, pointing to important roles of TNKS in this therapeutically interesting pathway and providing one of the few examples where the two tankyrases may have distinct functions.

This work is novel, timely and makes an important, long-awaited contribution to the field. The proteomics findings and TNKS/TNKS2 knockout cells will be an excellent resource. This impactful work will attract the attention from a wide readership studying the diverse functions of tankyrase.

Specific points:

(1) When discussing their proteomics work, the authors focus on a selection of tankyrase-regulated proteins, namely those containing potential tankyrase-binding motifs. A certain motif degeneracy means that these motifs can relatively easily be missed, as illustrated by the literature. (On that note, how were tankyrase-binding motifs identified? Were the candidate targets screened bioinformatically?) Moreover, there may be tankyrase targets that lack the known tankyrase-binding motif and interact with tankyrase either indirectly or via novel interaction modes, in addition to proteins that accumulate via mechanisms distinct from PAR-dependent ubiquitination. This would also make an important point for the Discussion section.

From the selected subset of proteins, the full complement of tankyrase-regulated processes cannot be fully appreciated. While Table S1 lists all identified 7254 proteins, it would be valuable to extend the discussion to those proteins whose abundance changes substantially but that do not have a clear tankyrase-binding motif. (Such motifs could be either absent or cryptic.) The authors could also point out those targets with the largest fold-changes in abundance in Figure 2D (around 2 dozens). It would certainly be appreciated to make the supplemental tables as accessible as possible and provide a table legend defining the various columns. The columns “Gene Name” and “Protein Name”, as included in Table S2, are very helpful, and could be added to Table S1 as well.

Upon surveying those proteins that accumulate upon loss of both TNKS and TNKS2, what is the estimated percentage of proteins that bear a detectable tankyrase-binding motif?

The tankyrase-binding motifs were identified by scanning each of the 287 proteins showing a significant change in abundance for a RxxxxG motif using the scanprosite tool and then rescanning those for a RxxG(P/A/C)xG site.

We have modified Table S1 and provide a figure legend. Sheet 1 lists the 7254 protein groups identified in the TMT analysis. Sheet 2 indicates the 608 proteins (out of 7254) showing a statistically significant change in abundance. They are indicated by (+) and we now include the gene name. Sheet 3 lists the 92 proteins with the greatest (greater than 1.5-fold) change in abundance, with the gene name now indicated. For the (27) proteins showing an increase in abundance, we now indicate the protein name and the RxxG(P/A/C)xG tankyrase binding site (if present).

We now add in the discussion on page 13: “Interestingly, there are proteins that show a significant increase in abundance but lack a RxxG(P/A/C)xG tankyrase binding site. These proteins may contain a cryptic motif. Alternatively, they could bind tankyrase through novel interaction modes or interact indirectly. In addition, proteins could accumulate by mechanisms that are distinct from PAR-dependent ubiquitination.”

Of the proteins that accumulate upon loss of both TNKS and TNKS2, 26% (72 out of 287) bear a RxxG(P/A/C)xG tankyrase binding site.

(2) An update for the Introduction (page 3): olaparib has been approved by the FDA and EMA for treatment of germline BRCA-mutated advanced ovarian cancer, in addition to ongoing clinical trials. Likewise, rucaparib has been FDA-approved.

We appreciate the update.

(3) Minor point, Introduction (page 3): the correct name of the SAM domain is “sterile alpha motif domain”.

We have corrected this.

(4) Introduction (page 3) and subsequent manuscript: the tankyrase-binding motif extends to a maximum of eight (rather than the minimal six) amino acids. While a rigid consensus probably “misses” degenerate motifs, reported consensus sequences, derived from experimental approaches and sequence analysis, include Arg-x-x-[small hydrophobic amino acids / Gly]-[Asp/Glu, in addition to a small selection of other tolerated amino acids]-Gly-[no Pro]-[Asp/Glu] (Guettler et al., 2011). While not every motif is eight amino acids long, it appears that the eighth position may be relevant to some of the novel tankyrase target candidates reported here, such as NOTCH1-4, which feature an aspartate residue at position eight, which almost certainly will contribute to binding. In Figure 2E, can the motif sequence shown be extended accordingly?

We now describe the 8 amino acid binding motif in the Introduction on page 3: “The tankyrase binding site recognized by the ARCs was initially identified as a six amino acid RxxPDG motif (Sbodio and Chi, 2002) that (through experimental approaches and sequence analysis) was extended to a maximum of eight amino acids: Rxx(small hydrophobic amino acids/G)(D/E, in addition to a small selection of other tolerated amino acids)G(no P)(D/E)(Guettler et al., 2011)”.

We now show the extended motif in Fig. 2E.

(5) Page 5, first paragraph: can you mention all KOs you generated in the first sentence?

We now mention all the clones in the first sentence on page 5: “We used CRISPR/Cas9 technology to generate tankyrase knockout (TNKS KO) human HEK293T cell lines; three TNKS1 KO; one TNKS2 KO; and one TNKS1/TNKS2 KO clones were generated (Fig. S1A-B).”

(6) Page 5, second paragraph and Discussion: an alternative interpretation of a requirement for both tankyrases is total tankyrase level (dosage). Where knockout of a single tankyrase leads to a phenotype (telomere cohesion, Notch localisation), it remains possible that these processes require a higher overall level of tankyrase, regardless of any potential differences between TNKS and TNKS2. In this case, the level of either tankyrase protein alone may not be sufficient. Differential phenotypes for single knockouts may simply reflect different expression levels of TNKS vs. TNKS2. In other words, losing the more abundant tankyrase protein will result in a more striking phenotype.

We feel that it is not a question of abundance particularly considering telomere cohesion, where we show that reintroduction (overexpression) of TNKS1 or TNKS2 was insufficient to rescue persistent telomere cohesion in the DKO; rescue required introduction of both TNKS1 and TNKS2 (Fig. S1F), consistent with each being required to resolve telomere cohesion. We now provide new data for spindle integrity function in Fig. S1G where we show that reintroduction (overexpression) of TNKS1 or TNKS2 was insufficient to rescue spindle defects in the DKO; rescue required introduction of both TNKS1 and TNKS2.

(7) Figure 2A: the effect on beta-catenin levels does not seem particularly striking (compare tubulin loading control and beta-catenin panel), although it is detectable. Can this blot and replicate blots be quantified?

Quantification of the blots is now indicated in Fig. 2A.

From the Methods section, it appears that total cell lysates were analysed, which would include membrane-associated beta-catenin. This may explain the subtle effect, given that the bulk of cellular beta-catenin is membrane-associated and not subject to regulation by Wnt/beta-catenin signalling. The authors may point out in the figure legend that total lysates were analysed.

We now state it in the figure legend.

(8) Page 8, paragraph 2: the term “in vivo” may not be appropriate for experiments in cell lines. Simply stating that the interaction of endogenous proteins was analysed would be sufficient.

We removed in vivo.

(9) Page 11, Discussion, paragraph 1: “heteropolymer” may be the more appropriate description, rather than “heterodimer”, since tankyrases can self-associate into heterotypic oligomers (De Rycker et al., 2004; Mariotti et al., 2016; Riccio et al., 2016).

We now added “.....heteropolymer, since tankyrases can self-associate into heterotypic oligomers (De Rycker et al., 2004; Mariotti et al., 2016; Riccio et al., 2016).”

(10) Minor point (typo), page 11, Discussion, paragraph 2: “grown” instead of “gown”

Corrected.

(11) Page 12, Discussion, paragraph 1: the actual reason for why Wnt/beta-catenin signalling is challenging to target may be its complexity and the redundancy of components, its wide involvement in adult tissue stem cell maintenance (giving rise to toxicity of pathway inhibitors) and the limited number of targetable components, rather than the fact that it is highly mutated in cancer.

We think that the pathway is difficult to target because the pathway is highly mutated in cancer – we now provide a reference that discusses this issue (Fearon, 2009).

In the same paragraph, the discussion that NKD2 induction may underlie the effect of JW74 on apoptosis is perhaps a bit too speculative. It might be removed or discussed as speculation, unless other data are available, but at least the cited paper does not appear to include such data.

We changed “perhaps” to “we speculate”.

(12) Minor point, figure 1, legend: HEK293T instead of Hek293T

Corrected.

(13) Figure 2, legend: the term “reduction” rather than “loss” of beta-catenin appears more appropriate. (See comments above on labelling panel D and displaying an extended Tankyrase-binding motif in panel E of figure 2.)

We changed “loss” to “reduction”.

(14) Figure 3D, legend: the term “plot” rather than “graphical representation” may be more appropriate.

We changed it to “plot”.

(15) Figure 5A, legend: can you define the abbreviation EFX?

The domain containing the EF-hand motif. We added it to the Figure legend.

(16) Figure 6, legend: can you define the abbreviations RAM and RBPJ (here and in the text)

RBPJ (recombining binding protein suppressor of hairless) and RAM (RPBJ association molecule) has been provided in the text and Figure legend.

and correct a typo (should read: “...tankyrase binding site across human Notch 1-4 ...”, delete “in”).

Corrected.

Also, the legend currently misses an explanation of the asterisks in panel E. In panel E, can the

heavy and light IgG chains be indicated and a typo (correct spelling “Coomassie”) be addressed?

The heavy and light IgG are indicated. Coomassie is corrected. The asterisks indicate the FlagNotch and MycTRF1 proteins bands. We indicate them now just in the Coomassie-stained gel for each lane and describe what they refer to in the figure legend.

The figure states that TRF1 is Myc-tagged while the legend refers to Flag-TRF1.

The legend was corrected to MycTRF1.

(17) Minor point: would the term “PARylation” be preferable to “PARsylation”? I.e., where does the “s” come from?

We agree with reviewer PARylation does seem to be used more in the current literature than PARsylation now so we changed it to PARylation. I think the “s” comes from ribosylate.

Reviewer #2 (Remarks to the Author):

To the authors:

The ms by Bhardwaj et al addresses novel target proteins for tankyrases, and also addresses the specific roles of tankyrase 1 and 2 via CRISPR/Cas9 targeting. They provide data on the specific roles of tankyrase 1 and 2 for telomere length, telomere cohesion and mitotic spindle integrity. They also conduct a proteomics analysis, using isomeric tag-based t-mass, which reveals novel target proteins, including the Notch 2 receptor. This prompted the authors do make a more deep-drilling effort into understanding the link between tankyrase and Notch (Notch2), and they provide evidence for direct interaction and that tankyrase ADP-ribosylates Notch2. Furthermore, the routing of Notch2 from the plasma membrane to the nucleus is affected by the tankyrase status. This represents a novel mode of posttranslational modification of the Notch intracellular domain, and may be important for future targeting of the Notch signaling pathway, or at least to understand possible adverse side-effect of tankyrase inhibitors.

Specific comments:

I think that the authors convincingly show that Notch2 is regulated by tankyrase (including mutating the putative binding site in Notch2). They also demonstrate that at least Notch1 and 3 are putative targets for tankyrase. It would be valuable to see whether these two Notch receptors also are de facto regulated by tankyrase. One reason for emphasizing on this experiment is that recent work on aPKC and PIM kinases have shown that they posttranslationally modify only specific Notch receptor paralogs, and potential differences in response to tankyrase among N1-N3 would aid in understanding possible paralog-specific functions.

For Notch 1, we provide new data showing that Notch1 is stabilized in HEK293T DKO cells compared to WT (Fig. S2A) and that introduction of tankyrase 1 and 2 into DKO cells rescues (reduces) Notch1 expression (Fig. S2B). For Notch3, we could not detect it in HEK293T cells, so we checked MCF7 cells (which were shown in the literature to express Notch 3) and we found that treatment with tankyrase inhibitor led to stabilization of Notch3 by immunoblot analysis (Fig. S2C) and on the cell surface shown by immunofluorescence analysis (Fig. S2D).

Figure 3A. How strong is the increase in Notch2 following tankyrase inhibitor exposure? Is it significant? The tubulin band is considerably stronger in the WT+Ti15 lane, and it would be good to see a densitometric scan + ratio to tubulin across all three lanes

We performed the scan and now provide the fold increase (2.1) in Fig. 3A. We also provide new data showing a 2.4 fold increase in Notch2 upon treatment with tankyrase inhibitor Ti8 (Fig. S2E).

Figure 3B. Why was not an inhibitor experiment (similar to Figure 3A) included also for Dicer, Chk2 and VAMP8?

Figure 3C. The change of cell line in Figure 3C (to HepG2 cells) and the use of different inhibitors only tested on Dicer is not explained in the text. Figure 3A-C thus gives a patchy impression, and it would be good to see that the effects are not restricted to specific cell lines/genes/inhibitors.

The purpose of the experiments in Fig. 3A and B was to show stabilization of the candidates in the DKO cells and provide the data for the graph Fig. 3D (now 3C). Some candidates were tested with inhibitors (Fig. 3A) and some not (Fig 3B). We were primarily concerned with validating the targets identified in our screen. For Fig. 3C (now 3D), we sought to analyze a different cell type from 293T (HepG2) using the same tankyrase specific inhibitor used with 293 cells (Ti8) as well as another commonly used general inhibitor (XAV939). We provide new data in Fig. 3C showing this for VAMP8 and Chk2 in addition to Dicer. VAMP8 shows stabilization, but not Chk2. We confirmed that Chk2 was not significantly stabilized by inhibitor treatment also in 293 cells. We now state in the Results on page 8: “Unlike the other targets Chk2 did not show stabilization with tankyrase inhibitor treatment.”

Figure 6B. The authors argue that there is more Notch2 ICD. By using the Cell signaling antibody it is not possible to distinguish Notch2 ICD from a membrane-tethered (NEXT) form of Notch2. There are now antibodies that identify the “neo-epitope” appearing in Notch2 ICD after S3 cleavage, and it would be important to use such an antibody to back up this claim.

We did not intend to argue that there was more Notch2 ICD. When we referred to Notch2-ICD, we were considering it to include the cleaved and uncleaved (membrane-tethered) form of the intracellular portion of Notch2. We have now corrected this throughout the text, stating that the cleaved form of the intracellular portion of Notch2 is the Notch2 ICD. As the Reviewer

states, the Cell Signaling antibody does not distinguish between these two forms. Upon the Reviewer's suggestion we obtained the anti-Notch2 (cleaved-Val1697) antibody that detects exclusively the cleaved form of Notch2 (Notch2-ICD) from Sigma. We provide new data showing that Notch-ICD is decreased upon treatment with tankyrase or gamma secretase inhibitors (Fig. S2F), consistent with the hypothesis that tankyrase (like gamma-secretase) is required for cleavage and release of Notch2 from the membrane. We are grateful to the reviewer for the suggestion and for helping us to clarify this issue in the Results and Discussion.

Figure 6F. The authors demonstrate a strong upregulation of plasma membrane-associated Notch2 staining in the tankyrase-deficient cells, which is interesting. As Notch receptors do not exclusively localize to the plasma membrane or nucleus, but a fair amount also resides in endocytic vesicles, it would be good to see if tankyrase also affects the localization to endosomes. This could be achieved by double-staining with markers for endosomes.

We appreciate the Reviewer's interest in the endosomal localization, however, we do not detect Notch2 in the endosomal compartment by IF and thus feel that a potential colocalization of a minor fraction would not be of significant impact in the context and focus of our work here.

To more precisely measure the amount of Notch2 receptor at the cell surface in response to tankyrase, a suggestion would be to use cell surface biotinylation for the Notch2 receptor. This would give a precise estimate of the extent of cell surface localization.

We show by immunofluorescence analysis that Notch2 is virtually undetectable in WT cells and that it is dramatically increased in DKO. We feel that a precise estimate of the extent will not extend the significance of this observation in the context and focus of our work here.

Figure 6F. Along the same lines, what would happen if a Notch2 construct similar to Notch1 Δ E, which is a truncated transmembrane form which does not require ligand activation but needs g-secretase processing, is used? Would such a construct also accumulate at the cell surface?

Yes, as we state in the Results section "The Notch2 staining pattern in the DKO cells was reminiscent of that observed for a Notch1 reporter allele (Notch1 Δ E-eGFP) in HeLa cells following treatment with the γ -secretase inhibitor DAPT; the Notch1 reporter accumulated at the plasma membrane rather than in the nucleus (Kramer et al., 2013)." This is what prompted us to treat cells with DAPT and we in fact show that all three treatments, TNKS DKO, TNKS inhibitor, and DAPT lead to a similar increase of Notch2 on the cell membrane (Fig. 6F and G).

An intriguing idea is whether tankyrase is "preloaded" on the intracellular domain of the Notch receptor already prior to cleavage of the Notch ICD. This could be tested by addressing whether tankyrase binds to Notch also when Notch S3-processing is blocked (by using g-secretase

inhibitors such as DAPT).

We provide new data in Fig. S2G. Immunoprecipitation analysis in DAPT-treated cells shows that tankyrase can bind to Notch2 when processing is blocked.

Figure 6H. The choice of nestin as a downstream Notch target gene is unconventional. There is indeed a report that nestin can be regulated by Notch in certain contexts, but there are a number of more established downstream genes, such as Hes or Hey genes, or NRARP. It would be good to see the nestin data complemented by these genes as well. In addition, the use of a direct downstream reporter construct, such as a multimerized CSL-binding site linked to luciferase, would provide a complementary type of analysis of downstream activation.

We selected Nestin because it was among the proteins that were significantly decreased in our TMT screen. To determine if the reduction in Nestin was really due to the tankyrase-Notch axis we measured Nestin levels in Control versus cells treated with tankyrase inhibitor Ti8 or the gamma-secretase inhibitor DAPT. We provide new data (Fig. 6I) to show that these treatments give a similar reduction in Nestin.

We agree it would be interesting to look at other downstream target genes such as Hes or Hey. We tested these using our HEK293T cell system with DKO cells or DAPT treatment as a control, but were unable to see reproducible effects - even with DAPT. This more extensive type of analysis would likely require establishing a ligand-inducible system, perhaps in another cell type. We feel that this is beyond the scope of the manuscript.

Minor comments:

Discussion: The discussion on lethality in vivo for the double tankyrase OK embryos versus survival of the HEK293 cells in which T1 and T2 are removed seems a bit naïve – it really takes a lot to kill an established cell line such as HEK293.

We feel that it is still worth pointing out that the DKO is lethal in mouse. We highlight the difference between requirements for the early development of an organism versus requirements for cancer cells in culture (in agreement with the reviewer).

Mice lacking both T1 and T2 are embryonically lethal – is there by any chance phenotypes which resemble increased Notch signaling. A good experiment would be to look for higher levels of Notch2 in the double knockout mouse embryos (although I realize that it is a daunting experiment if the authors do not have access to the mice, or can obtain embryo material from another group).

We agree it would be an interesting experiment. My lab contributed to the double knockout mouse work published in 2008. The E10 DKO embryos were difficult to obtain and severely

deteriorated. We no longer have the mice or embryonic material.

The authors cite the reviews by Carrieri and Dale 2016 and Nowell and Radtke 2017 for Notch oncogene reviews. The Nowell/Radtke review basically addresses the tumor suppressor function and a more recent, and general, review would be Aster, Pear, Blacklow Ann. Rev Pathol 2017.

We added the Aster review.

Reviewer #3 (Remarks to the Author):

In the manuscript entitled “Whole proteome analysis of human tankyrase knockout cells reveals new targets of tankyrase-mediated degradation”, the authors established a TNKS1/2 double knockout cells in HEK293T cell line by using CRISPR/Cas9, and systematically analyzed the degrading substrates for TNKS1/2 via TMT-based proteomics. Through it, they not only validated previously identified TNSK1/2 substrates for degradation, but also uncovered novel candidate substrates in different pathways or cellular functions (WNT, miRNA, and NOTCH). Overall, this study is very interesting and provides a useful resource for the field to further characterize the cellular functions of TNKS1/2.

Major points:

1. In Fig 1, the authors claimed that they generated TNKS1/2 single and double KO cells, and functionally validated these cells. In addition, they took advantage of these established KO cells to compare the roles between TNKS1 and TNKS2 by using telomere and mitosis-related assays. It seems that TNKS1 and 2 showed redundant functions in all these assays (Fig1C, 1F and 1H). If the authors really want to claim that TNKS1 and TNKS2 are both required for some cellular functions (i.e. telomere cohesion and mitotic spindle integrity), they at least need to 1) perform the significance analysis (p value) between DKO cell and single KO cells in Fig. 1C and Fig.1H; and 2) show rescue data in DKO cells by using TNKS1 or TNKS2 alone or in a combination.

We think our data are consistent with redundant roles for TNKS1 and TNKS2 in telomere length maintenance and distinct roles in resolution of telomere cohesion and spindle integrity. We showed in Fig. 1 that the same TNKS1 KO or TNKS2 KO cells that maintain telomere length (Fig. 1F) cannot resolve telomere cohesion (Fig. 1C) or maintain spindle integrity (Fig. 1H).

Regarding rescue data: For telomere length we showed that introduction of either TNKS1 or TNKS2 into DKO cells was sufficient to rescue telomere length maintenance (Fig. 1SE), indicating that either could provide the telomere length function. For resolution of telomere cohesion, we showed that introduction of either TNKS1 or TNKS2 into DKO cells was not sufficient to rescue resolution of cohesion; this required introduction of both TNKS1 and TNKS2 (Fig. S1F),

indicating that both are required to resolve cohesion. For spindle integrity, we provide new data (Fig. S1G) showing that introduction of either TNKS1 or TNKS2 into DKO cells was not sufficient to rescue spindle defects; this required introduction of both TNKS1 and TNKS2, indicating that both are required for spindle integrity.

For statistical analysis we provided statistical analysis in the paper showing significant statistical difference between the single and DKO cell lines versus wild type. We were not claiming that there was statistical significance between the DKO and single KOs. This is difficult to show in all cases due to the upper limits of the biological assays we use. We feel that the rescue analysis makes the point that either TNKS1 or TNKS2 can support telomere length maintenance, but both are required for resolution of telomere cohesion and spindle integrity. We nonetheless performed the statistical analysis requested and provide it here for the reviewer. Statistical analysis for telomere cohesion shows for DKO versus TNKS1 ($p=0.082$) (ns) and DKO versus TNKS2 ($p=0.039$) (*). For spindle integrity, statistical analysis for aberrant spindle shows for DKO versus TNKS1 ($p=0.052$) (ns) and for DKO versus TNKS2 ($p=0.20$) (ns). For spindle integrity, statistical analysis for multipolarity shows for DKO versus TNKS1 ($p=0.022$) (*) and for DKO versus TNKS2 ($p=0.022$) (*).

2. As for the TMT-MS validation data in Fig3, the authors used tankyrase inhibitors (compounds 8 and 15) as control to validate the candidate expression in DKO cells. Since tankyrase inhibitors target tankyrase-enzyme activity to stabilize the substrates, the DKO should have at least the close or even higher effect than the inhibitors to stabilize the substrates. In that case, why did AMOT and NKD2 have higher expressions in inhibitor-treated condition than in the DKO cells? As for Fig3A, inhibitor treatment should also be performed in DKO cells to serve as a control.

We would not anticipate that the inhibitors will have the same or less of an effect as DKO. Inhibitor treatment leads to dramatic stabilization of TNKS (see Fig. 3A top panel right-most lane) that could lead to aggregation. Regarding using DKO as a control for inhibitor treatment, we provide new data in Fig. 3E where we performed inhibitor treatment of WT cells side-by-side with DKO cells for the established target AMOT and the new target Notch2 and show that while the inhibitor stabilizes AMOT and Notch2 in control cells it has no effect in TNKS DKO cells.

3. In Fig4A, the authors performed rescue experiments in DKO cells, which is quite important and helpful to validate their DKO cells. Since tankyrase PAR activity is required to target substrate for degradation, it will be great if a PAR-deficient mutant of TNKS could be included here as a control.

We agree with the reviewer and we tried these experiments, but did not see consistent results. In some cases PARP-dead TNKS influenced protein stability similar to WT tankyrase. We checked the literature, but most papers used tankyrase inhibitors rather than PARP dead tankyrase to show a dependence on catalytic activity. Overexpressed PARP-dead tankyrase

could serve a scaffolding role, recruiting RNF146 (which can bind to tankyrase directly independent of PARylation; DaRosa et al., 2015) and tankyrase-binding proteins to promote degradation. In this context minor residual catalytic activity in the PAR-dead allele could also contribute. We feel that our demonstration that the targets that are stabilized in DKO cells are also stabilized in WT cells treated with inhibitor, indicates a dependence on PARP activity.

4. In Fig6, the authors validated Notch receptors as targets for TNKS1/2-mediated degradation. The biochemical experiments are fine, but I am confused about the notch-related functional studies presented here. Based on the provided data, it seems that TNKS1/2 can not only degrade Notch2 but also activate Notch signaling by enhancing Notch2 nuclear localization. This discrepancy needs clarification.

Compared to DKO and Ti8-treated cells, Notch2 seems to be undetected in WT cells. Based on the figures provided (Fig6F and 6G), I could see more nuclear staining of Notch2 in DKO and Ti8-treated cells compared to that in WT cells.

Our data indicate that Notch2 is increased on the plasma membrane; it not increased in the nucleus in DKO (Fig. 6F) or Ti8- and DAPT-treated cells (Fig. 6G). The reviewer may have been misled by the DAPI merge and the non-specific background in the HeLa cells in 6G. We provide the reviewer with the unmerged Notch2 images (at the end of this document). For Fig. 1F, we do not see increased nuclear stain in the HEK293T DKO cells compared to WT. For Fig. 6G, these are HeLa cells. These cells show a high nuclear background (likely non-specific) with the Notch2 antibody even in the untreated control cells. This nuclear background stays the same in Ti8 and DAPT treated cells; it does not increase.

As for Fig6H, there is no direct evidence provided here to show that downregulation of Nestin transcription is through TNKS1/2-mediated alteration of Notch2 (degradation? membrane retention?).

We have now performed this analysis on cells treated with gamma-secretase inhibitor DAPT and we see a similar reduction in Nestin mRNA levels as with Ti8 (or DKO). (new Fig. 6I). This, combined with our observation in Fig. 6G that DAPT and leads to a similar accumulation of Notch2 on the plasma membrane as Ti8 (or DKO), suggests that they might be acting in the same pathway to increase the membrane bound form of Notch2.

Minor points:

1. In Fig1G and 1H, if >40% DKO cells have mitotic defects, how would the authors establish and obtain the DKO cells? Is there any cell proliferation difference between WT, single and double KO cells?

We agree with the reviewer, it is surprising that the cells grow so well. As Reviewer 1 points out above "... it really takes a lot to kill an established cell line such as HEK293." We do not see dramatic proliferation differences between WT, single, and double KO cells.

2. In Fig2A, the tubulin blot seems also decreased in KO and DKO cells.

We have now quantified the blots relative to tubulin in Fig. 2A. We show a clear increase (2.5 fold) in Axin1 and decrease (0.3 fold) in beta-catenin in DKO versus WT cells.

3. In Fig6C, what is "tx"? I guess that it is short for "transfected".

Yes.

4. In Fig6G, can γ -secretase inhibitor stabilize Notch2? Its treatment seems to have the similar effect on Notch2 protein level to Ti8.

We provide new data in Fig. S2E. Immunoblot analysis of DAPT treated cells shows an increase in Notch2 levels, similar to but not as high as Ti8.

Notch2 only staining for Fig. 6F and 6G.

REVIEWERS' COMMENTS:

Reviewer #1 (Remarks to the Author):

Most of the points I raised have been addressed. As I said in my original review, this is a very interesting study.

There are only a few minor points that require revision/clarification:

- Although acknowledged out in the rebuttal, the introduction (page 3, first paragraph) has not been updated to reflect that PARP inhibitors are now in clinical use. The papers cited are from 2014, and the authors may want to update their manuscript accordingly.
- On page 5, first paragraph, the authors state that TNKS and TNKS2 cannot compensate for each other (for certain functions). Perhaps it is best to point out that this is at endogenous levels. The compensation experiment, which supports the statement, is not described until the end of paragraph 2 on page 5. For clarity, can the rescue experiment for the telomere cohesion defect be moved up, and can the authors point out that the experiment entails overexpression (here and for other rescue experiments)? The level of overexpression (x-fold) could also be referred to, in all rescue experiments described in the manuscript. The overexpression is indeed a strong argument against the total dose of tankyrase 1+2 being more important than the proposed functional differences between the tankyrases or some putative mechanism that requires collaboration of tankyrase 1 and 2. An extended elaboration of this point would add clarity to the manuscript.
- For Figure 2A, although acknowledged in the rebuttal, the change in the figure legend has not been made.
- With an acidic residue at the 8th position of the tankyrase binding motif, Notch almost certainly has an 8-amino-acid TBM. This could be pointed out in Figure 6A.
- page 11, first paragraph of Discussion: there is another occurrence of the term "in vivo" used for cell culture experiments. Can this be changed, please?
- The observation that some proteins increase in abundance and an almost equal number of proteins decrease in abundance upon loss of both tankyrases is interesting. Can this be discussed further in the Discussion?
- Can Western blots confirming RNF146 knockdown be included for the experiments shown in Figures 5B and 6B?
- Figures 5C and D show reduced interaction of TBM-mutated NKD2 with tankyrase, but the text states that the interaction is "abolished". "Strongly reduced" appears more appropriate to describe the data.
- At the end of the second paragraph on page 9, the authors state that "... Notch1 is [...] rescued by reintroduction of tankyrase 1 and 2." This wording is somewhat ambiguous, and it may be better to state that low levels of Notch 1 are restored by reintroduction of the two tankyrases.
- page 12, top: the 74 tankyrase binding sites should be referred to as "putative" since the majority are not confirmed.
- page 12, third paragraph: it is confusing that two consecutive sentences state that the "intracellular portion of Notch2 was stabilized" upon tankyrase knockout and that "increased

Notch2 localized to the plasma membrane, suggesting that it was the uncleaved, membrane-bound form of the intracellular portion that was stabilized [...]" I am confused as to whether the cleaved or the uncleaved protein is stabilized. Can the authors please clarify?

- Fearon, 2009, discusses the challenge of identifying druggable nodes in a complex pathway, and tankyrase represents such a node. In the Discussion on page 12, the authors could highlight the challenge of the limited number of targetable pathway components, which is one of many major hurdles in targeting the Wnt pathway. This is one reason for why tankyrase attracted so much attention.

Reviewer #2 (Remarks to the Author):

The authors have addressed most points of critique and for the points not addressed, I accept their explanations for not conducting the suggested experiments.

Reviewer #3 (Remarks to the Author):

Most points I raised earlier have been addressed satisfactorily. I thank the authors for having considered seriously my comments on their original submission.

Response to Reviewers

Reviewer #1 (Remarks to the Author):

Most of the points I raised have been addressed. As I said in my original review, this is a very interesting study.

There are only a few minor points that require revision/clarification:

- Although acknowledged out in the rebuttal, the introduction (page 3, first paragraph) has not been updated to reflect that PARP inhibitors are now in clinical use. The papers cited are from 2014, and the authors may want to update their manuscript accordingly.

We now cite a 2017 review (Bitler et al) on PARP inhibitors in clinical use on page 3.

- On page 5, first paragraph, the authors state that TNKS and TNKS2 cannot compensate for each other (for certain functions). Perhaps it is best to point out that this is at endogenous levels. The compensation experiment, which supports the statement, is not described until the end of paragraph 2 on page 5. For clarity, can the rescue experiment for the telomere cohesion defect be moved up, and can the authors point out that the experiment entails overexpression (here and for other rescue experiments)? The level of overexpression (x-fold) could also be referred to, in all rescue experiments described in the manuscript. The overexpression is indeed a strong argument against the total dose of tankyrase 1+2 being more important than the proposed functional differences between the tankyrases or some putative mechanism that requires collaboration of tankyrase 1 and 2. An extended elaboration of this point would add clarity to the manuscript.

For the Results on page 5 and 6:

We provide immunoblots for the rescued cell lines in a new Supplementary Figure 2a showing that the proteins are highly overexpressed, greater than 10-fold more than endogenous tankyrases. These cell lines were used for all the rescue experiments in the paper. We moved old Fig. S1F (the cohesion rescue experiment) up to Supplementary Figure 2b to discuss it immediately after the cohesion analysis.

We now state on page 5 first paragraph, that for resolution of telomere cohesion: “at endogenous levels tankyrase 1 and 2 are each required and one cannot compensate for the other. To perform rescue analysis, we generated stable DKO cell lines overexpressing tankyrase 1, tankyrase 2, or both at levels greater than 10-fold relative to the endogenous proteins (Supplementary Figure 2a). Overexpression of tankyrase 1 or 2 was unable to rescue persistent telomere cohesion; this required overexpression of both tankyrase 1 and 2 (Supplementary Figure 2b), consistent with each being required.

We now state on page 5 second paragraph, that for telomere length: “Overexpression (greater than 10-fold) of either tankyrase 1 or 2 in DKO cells rescued telomere shortening (Supplementary Figure 2e), consistent with the observation that either tankyrase 1 or 2 can maintain telomere length”

We now state on page 6 first paragraph, that for mitotic spindle integrity: “Overexpression of tankyrase 1 or 2 at levels greater than 10-fold relative to the endogenous proteins was unable to rescue spindle defects; this required overexpression of both tankyrase 1 and 2 (Supplementary Figure 2f), consistent with each being required.”

For the Discussion on page 12 first paragraph:

We now state: “Our observation, that overexpression of tankyrase 1 or 2 at levels greater than 10-fold relative to the endogenous proteins was unable to rescue persistent cohesion or mitotic spindle integrity, argues against the idea that the total amount of tankyrase 1 and 2 is required for function and

instead suggests a functional difference between tankyrases and/or a putative mechanism that requires collaboration.”

- For Figure 2A, although acknowledged in the rebuttal, the change in the figure legend has not been made.

We now state in Figure Legend 2a “total cell lysates”.

- With an acidic residue at the 8th position of the tankyrase binding motif, Notch almost certainly has an 8-amino-acid TBM. This could be pointed out in Figure 6A.

We now state in Figure Legend 6A that “The acidic residue at the 8th position of the tankyrase binding motif indicates an 8-amino-acid motif.”

- page 11, first paragraph of Discussion: there is another occurrence of the term "in vivo" used for cell culture experiments. Can this be changed, please?

We changed in vivo to “in living cells”

- The observation that some proteins increase in abundance and an almost equal number of proteins decrease in abundance upon loss of both tankyrases is interesting. Can this be discussed further in the Discussion?

We modified the third paragraph of the Discussion (shown below, indicated in black):

We performed an unbiased quantitative proteomic screen in WT versus DKO cells and identified 74 putative tankyrase-binding site containing candidate targets of tankyrase-mediated degradation. Six of the 74 were previously identified and we confirmed seven novel candidates, thereby giving confidence in our strategy for finding targets of tankyrase-mediated degradation. While some of the candidates could be pooled into small functional groups, many represented an array of unique functions. This is perhaps not surprising, as tankyrases localizes throughout the cell to multiple compartments and exhibit a broad range of functions. The identification of novel targets will likely provide insights into tankyrase function and will also reveal new roles. **While we focused on the 74 described above, our screen yielded a total of 608 proteins with a significant change in abundance; some proteins increased, while a similar number decreased. A number of proteins that show a significant increase lack a RxxG(P/A/C)xG tankyrase binding site. These proteins may contain a cryptic motif. Alternatively, they could bind tankyrase through novel interaction modes or interact indirectly. In addition, proteins could accumulate by mechanisms that are distinct from PAR-dependent ubiquitylation. Regarding proteins that decreased in abundance, some (such as Nestin) may result indirectly from proteins stabilized in the absence of tankyrase. Bioinformatic analysis for protein-protein interactions among the 608 proteins will likely identify potential connections that can be experimentally validated.**

- Can Western blots confirming RNF146 knockdown be included for the experiments shown in Figures 5B and 6B?

We now provide new western blot data in Fig. 5b confirming the RNF146 knockdown for the experiments shown in Figures 5b and 6b.

- Figures 5C and D show reduced interaction of TBM-mutated NKD2 with tankyrase, but the text states that the interaction is "abolished". "Strongly reduced" appears more appropriate to describe the data.

We now state on page 9:

“As shown in Fig. 5c, GFP-NKD2 WT coimmunoprecipitated with endogenous TNKS1, whereas interaction with the GFP-NKD2 mutant was strongly reduced. Conversely, endogenous TNKS1 coimmunoprecipitated with GFP-NKD2 WT, whereas interaction with the GFP-NKD2 mutant was strongly reduced. (Fig. 5d).”

- At the end of the second paragraph on page 9, the authors state that "... Notch1 is [...] rescued by reintroduction of tankyrase 1 and 2." This wording is somewhat ambiguous, and it may be better to state that low levels of Notch 1 are restored by reintroduction of the two tankyrases.

We now state on page 10 first paragraph, that “the low levels of Notch 1 are restored by reintroduction of tankyrase 1 and 2”

- page 12, top: the 74 tankyrase binding sites should be referred to as "putative" since the majority are not confirmed.

We changed it (on page 13 first paragraph) to “74 putative tankyrase-binding site containing....”

- page 12, third paragraph: it is confusing that two consecutive sentences state that the "intracellular portion of Notch2 was stabilized" upon tankyrase knockout and that "increased Notch2 localized to the plasma membrane, suggesting that it was the uncleaved, membrane-bound form of the intracellular portion that was stabilized [...]" I am confused as to whether the cleaved or the uncleaved protein is stabilized. Can the authors please clarify?

The uncleaved protein is stabilized. We state this clearly in the results on page 11 first paragraph:

“We next sought to determine if it was the cleaved ICD or uncleaved membrane-bound form of Notch2 that was increased following inhibition of tankyrases. Our immunoblot analysis thus far (using antibody directed against the C-terminus of Notch2) did not distinguish between the two forms, since they differ only slightly in molecular weight. We thus performed the analysis using antibody that detects only the cleaved (Val1697) form of Notch2. As shown in Fig. S2F, we observed a reduction in cleaved Notch2 upon Ti8 or DAPT treatment, indicating that the observed increase in Notch2 upon tankyrase inhibition is due to an increase in the membrane-bound uncleaved form.”

- Fearon, 2009, discusses the challenge of identifying druggable nodes in a complex pathway, and tankyrase represents such a node. In the Discussion on page 12, the authors could highlight the challenge of the limited number of targetable pathway components, which is one of many major hurdles in targeting the Wnt pathway. This is one reason for why tankyrase attracted so much attention.

We now state in the Discussion on page 13 second paragraph:

“Considering the limited number of known targetable enzymes in Wnt/ β -catenin signaling, the identification of tankyrase as a druggable node in the pathway opened up new possibilities.”

Reviewer #2 (Remarks to the Author):

The authors have addressed most points of critique and for the points not addressed, I accept their

explanations for not conducting the suggested experiments.

Reviewer #3 (Remarks to the Author):

Most points I raised earlier have been addressed satisfactorily. I thank the authors for having considered seriously my comments on their original submission.